# The use of radiocarbon [14]C to constrain carbon dynamics in the soil module of the land surface model ORCHIDEE (SVN r5165)

**Marwa Tifafi[1], Marta Camino-Serrano[2,3], Christine Hatté[1], Hector Morras[4], Lucas Moretti[5], Sebastián Barbaro[5], Sophie Cornu[6], Bertrand Guenet[1]**

[1]Laboratoire des Sciences du Climat et de l'Environnement, LSCE/IPSL, CEA-CNRS-UVSQ, Université Paris-Saclay, F-91191 Gif-sur-Yvette, France.
[2]CREAF, Cerdanyola del Vallès, 08193, Catalonia, Spain
[3]CSIC, Global Ecology Unit CREAF-CSIC-UAB, Bellaterra 08193, Catalonia, Spain
[4]INTA-CIRN, Instituto de Suelos, 1712 Castelar, Buenos Aires, Argentina
[5] INTA-EEA Cerro Azul, 3313 Cerro Azul, Misiones, Argentina
[6]Aix Marseille Univ, CNRS, IRD, INRA, Coll France, CEREGE, Aix-en-Provence, France

Corresponding authors: Marwa Tifafi (marwa.tifafi@lsce.ipsl.fr)

**Abstract.** Despite the importance of soil as a large component of the terrestrial ecosystems, the soil compartments are not well represented in the Land Surface Models (LSMs). Indeed, soils in current LSMs are generally represented based on a very simplified schema that can induce a misrepresentation of the deep dynamics of soil carbon. Here, we present a new version of the Institut Pierre Simon Laplace (IPSL) Land Surface Model called ORCHIDEE-SOM (ORganizing Carbon and Hydrology in Dynamic EcosystEms-Soil Organic Matter), incorporating the [14]C dynamic in the soil. ORCHIDEE-SOM first simulates soil carbon dynamics for different layers, down to 2 m depth. Second, concentration of dissolved organic carbon and its transport are modeled. Finally, soil organic carbon decomposition is considered taking into account the priming effect.

After implementing [14]C in the soil module of the model, we evaluated model outputs against observations of soil organic carbon and modern [14]C fraction ($F^{14}C$) for different sites with different characteristics. The model managed to reproduce the soil organic carbon stocks and the $F^{14}C$ along the vertical profiles for the sites examined. However, an overestimation of the total carbon stock was noted, primarily on the surface layer. Due to [14]C, it is possible to probe carbon age in the soil, which was found to underestimated. Thereafter, two different tests on this new version have been established. The first was to increase carbon residence time of the passive pool and decrease the flux from the slow pool to the passive pool. The second was to establish an equation of diffusion, initially constant throughout the profile, making it vary exponentially as a function of depth. The first modifications did not improve the capacity of the model to reproduce observations whereas the second test improved both estimation of surface soil carbon stock as well as soil carbon age. This demonstrates that we should focus more on vertical variation of soil parameters as a function of depth, in order to upgrade the representation of global carbon cycle in LSMs, thereby helping to improve predictions of the of soil organic carbon to environmental changes.

## 1 Introduction

The complexity of the mechanisms involved in controlling soil activity (Jastrow et al., 2007) and therefore the carbon flux from the soil to the atmosphere makes predicting the response of these systems to climate change extremely complex. Thus our ability to predict future changes in carbon stocks in soils using global climate models is currently heavily criticized (Todd-Brown et al., 2013; Wieder et al., 2013). Indeed, Earth System Models (ESMs) are increasingly used today in order to predict the future evolution of the climate. For instance, results of a set of ESMs are taken into account within the Intergovernmental Panel on Climate Change (IPCC) (Taylor et al., 2012) for assessment of the impacts of climate change and design of mitigation strategies. Hence, their predictions need to be as accurate as possible. These models represent the physical, chemical and biological processes within and between the atmosphere, ocean and terrestrial biosphere. They allow us to follow and understand both the effect of the climate on carbon storage and vice versa. However, ESMs are continuously under development and some key processes in the global carbon cycle are still missing or not represented with the necessary details. One of the components of an ESM is the land surface model (LSM). This component primarily manages the carbon cycle, energy and water on land and simulates the carbon exchange between the land surface and the atmosphere, namely the gross primary production (GPP), the autrophic and heterotrophic respiration.

Despite the importance of soils as a large component of the global carbon storage, soil compartments are not well represented in LSMs (Todd-Brown et al., 2013). Indeed, carbon dynamics in soil described in LSMs are based on the "Century" (Parton et al., 1987) or Roth-C models (Coleman et al., 1997) where soil carbon is represented as several pools with different turnover rates for each pool. Carbon is decomposed in each pool, one part of which is then transferred from one pool to another and the other part is lost through heterotrophic respiration. In addition, soils are generally represented as a single-layer box in LSMs that do not take into account the evolution and variation of soil organic processes as a function of depth (Todd-Brown et al., 2013).

One way to reconcile this simplified representation of carbon dynamics of the models with the complexity of the data collected in the field is to integrate isotopic tracers into the models themselves and thus facilitate the comparison between model outputs and data (He et al., 2016). Moreover , thanks to an additive constraints on the model structure, this may improve the model performances. For instance, radiocarbon is an important tool for studying the dynamics of soil organic matter (Trumbore, 2000). Indeed, $^{14}$C data acquired from soil organic matter provide complementary information on the dynamics (temporal dimension) of soil organic matter. This tracer has the major advantage of being integrator of carbon dynamics on long time scales (a few decades to several centuries). It is therefore a very powerful tool to constrain conceptual schemes that may not be directly compared to variables measured in the field (Elliott et al., 1996). Different authors have succesfully implemented radiocarbon in soil models and were able to clearly show that the introduction of pools with turnover time of thousands of year were unnecessarry to fit radiocarbon data (Ahrens et al., 2015) whereas Braakhekke et al., (2014) showed that after a reparameterization of the models based on radiocarbon data the prediction of their model was quite different with more carbon in top soil and less in deep soil compared to the model without radiocarbon.

Radiocarbon is produced naturally at a constant rate in the upper atmosphere through bombardment of cosmic rays. It thus provides information on the dynamics of organic matter that has been stabilized by interaction with mineral surfaces and stored long enough for significant radioactive decay (Trumbore, 2000), as the half-life of $^{14}$C is about 5730 years. We must also take into account radiocarbon produced during atmospheric tests of thermonuclear weapons in the early sixties (Delibrias et al., 1964; Hua et al., 2013). Atmospheric bomb testing in the late 1950s and early 1960s lead to an abrupt doubling of atmospheric $^{14}$C concentration in a span of 2-3 years. Through exchange with ocean and terrestrial reservoirs, it has decreased but still remains above the natural background. As with any other carbon isotopes, this $^{14}$C was metabolized by the vegetation and transferred to soil. By measuring $^{14}$C activity of a soil sample, it is possible to evaluate the amount of carbon introduced into the soil since the 1960s (Balesdent and Guillet, 1982; Scharpenseel and Schiffmann, 1977).

In this study, we present a new version of the IPSL-Land Surface Model called ORCHIDEE-SOM incorporating $^{14}$C dynamics in the soil. Thanks to this tracer, we can evaluate the SOC dynamics, in particular by looking at the $^{14}$C peak produced by atmospheric weapons testing and observed in the soils at four different sites having different biomes.

## 2 Materials and methods

### 2.1 ORCHIDEE-SOM overview

ORCHIDEE is the Land Surface Model of the IPSL Earth System Model (Krinner et al., 2005). It is composed of three different modules. First, SECHIBA (Ducoudré et al., 1993; Rosnay and Polcher, 1998), the surface-vegetation-atmosphere transfer scheme, describes the soil water budget and energy and water exchanges. The time step of this module is 30 min. Second, the module of the vegetation dynamics has been taken from the dynamic global vegetation model LPJ (Sitch et al., 2003). The time step of this module is one year. Finally, the STOMATE (Saclay Toulouse Orsay Model for the Analysis of Terrestrial Ecosystems) module simulates vegetation phenology and carbon dynamics with a time step of one day.

ORCHIDEE can be run coupled to a global circulation model where the boundary conditions of the model are provided by the atmospheric modules (temperature, precipitation, atmospheric $CO_2$ concentration, etc.). In return ORCHIDEE provides the land surface carbon, energy and water fluxes. However, since our study focuses on changes in the land surface rather than on the interaction with climate, we ran ORCHIDEE in the off-line configuration. In this case, atmospheric conditions such as temperature, humidity and wind are read from a meteorological dataset. The climate data CRUNCEP used for our study (6-hourly climate data over several years) were obtained from the combination of two existing datasets: the Climate Research Unit (CRU) (Mitchell et al., 2004) and the National Centers for Environmental Prediction (NCEP) (Kalnay et al., 1996).

Our starting point is a ORCHIDEE-SOM version based on the SVN r3340 (Krinner et al., 2005), which is presented in detail in Camino-Serrano et al. (2017). Figure 1 represents how the soil is described in this new version. Indeed, the major particularity of ORCHIDEE-SOM is that it simulates the dynamics of soil carbon for eleven layers from the surface to two

meters depth. First, litter is divided into four pools: metabolic or structural litter pools which
can be found below or aboveground. Only the belowground litter is modeled on eleven levels,
from surface to 2 m depth, as the aboveground litter layer has a fixed thickness of 10 mm.
Second, SOC is divided into three pools (active, passive and slow), following Parton et al.
(1988), which differ in their turnover rates and which are discretized into 11 layers up to a
depth of two meters. Then, dissolved organic carbon (DOC) is represented as two pools and
also discretized over 11 layers up to a depth of two meters: labile DOC has a high
decomposition rate and recalcitrant DOC has a low decomposition rate (Camino-Serrano et
al., 2018). Finally, another particularity of this version of ORCHIDEE-SOM is that the SOC
decomposition is modified to account for the priming effect following Guenet et al. (2016).
Briefly, priming is described following equation 1.
$$\frac{\partial SOC_{i,z}}{\partial t} = DOC_{Recycled,i,j}(t) - k_{SOC,i} \times (1 - e^{-c \times LOC_z(t)}) \times SOC(t)_{i,z} \times \theta(t) \times \tau(t) \qquad (1)$$
with $DOC_{recycled}$ being the unrespired DOC that is redistributed into the pool *i* considered for
each soil layer *z* in g C m$^{-2}$ days$^{-1}$, $k_{SOC}$ being a SOC decomposition rate constant (days$^{-1}$), and
*LOC* being the stock of labile organic C defined as the sum of the C pools with a higher
decomposition rate than the pool considered within each soil layer *z*. We therefore considered
that for the active carbon pool LOC is the litter and DOC, but for the slow carbon pool LOC
is the sum of the litter, DOC and so on. Finally, *c* is a parameter controlling the impact of the
*LOC* pool on the *SOC* mineralization rate, i.e., the priming effect. The equation was
parameterized based on soil incubations data and evaluated over litter manipulation
experiments (Guenet et al. 2016).
Since the soil profile is divided into 11 layers, SOC and DOC transport following the
diffusion must also be described. SOC diffusion is actually a representation of bioturbation
processes (animal and plant activity), whereas DOC relies more on non-biological diffusion.
Both diffuse through concentration gradients.
This is represented using the Fick's law (Braakhekke et al., 2011; Elzein and Balesdent, 1995;
O'Brien and Stout, 1978; Wynn et al., 2005):
$$F_D = -D * \frac{\partial^2 C}{\partial z^2} \qquad (2)$$
Where $F_D$ is the flux of carbon transported by diffusion in g C m$^{-3}$ day$^{-1}$, *D* is the diffusion
coefficient (m$^2$ day$^{-1}$) and *C* is the amount of carbon in the pool (DOC or SOC) subject to
transport (g C m$^{-3}$). The diffusion coefficient is assumed to be constant across the soil profile
in ORCHIDEE-SOM but the diffusion parameters (D) used in the equations for SOC and
DOC can differ. All the transport processes goes up to two meters, corresponding to the soil
depth fixed in the model. For DOC, at two meters the DOC can be exported through drainage.
**2.2 ORCHIDEE-SOM-$^{14}$C**
In ORCHIDEE-SOM, the different compartments (soil carbon input, litter, SOC, DOC and
heterotrophic respiration) are presented as a matrix with a single dimension referring to the
total carbon. In order to introduce the $^{14}$C, a new dimension has been added to all the
variables cited above. Thus, all processes that apply to the total soil carbon are now also
represented for $^{14}$C. We label this new version including $^{14}$C as ORCHIDEE-SOM-$^{14}$C.
Several ways of reporting $^{14}C$ activity levels are available. We chose to use the *fraction*
*modern*, with the $F^{14}C$ symbol as advocated by Reimer et al. ( 2004) rather than absolute
concentration of $^{14}C$ (reported as Bq).
$$F^{14}C = \left(\frac{A_S}{0.95\,A_{OX1}}\right) * \left(\frac{0.975}{0.981}\right)^2 * \left[\left(1 + \frac{\delta^{13}C_{OX1}}{1000}\right) \Big/ \left(1 + \frac{\delta^{13}C_S}{1000}\right)\right]^2 \quad (3)$$
with $A = {}^{14}C/{}^{12}C$, S for sample, OX1 for Oxalic Acid 1, the $^{14}C$ international standard.
$F^{14}C$ is twice normalized: i) it takes into account isotopic fractionation by being normalized to
a $\delta^{13}C = -25‰$, and ii) it corresponds to a deviation towards an international standard (i.e.
95% of OX1 as measured in 1950 – (Stuiver and Polach, 1977)). By propagating $F^{14}C$ from
atmosphere at the origin of vegetal photosynthesis to soil respired $CO_2$, there is no need to
focus on $^{13}C$ isotopic fractionation all along the organic matter mineralization with $F^{14}C$.
To make the reading of the paper easier, we will further express $F^{14}C$ as $F^{14}C = A_{sample}/A_{ref}$
with $A_{sample}$ being the A of the measured (or modeled) data and $A_{ref}$ an international reference.
Normalizations are included in $A_{ref}$ and $F^{14}C$ will be written as $F^{14}$ to simplify notation
involving superscripts and subscripts.
Since we focus on SOC dynamics, we did not include the $^{14}C$ in the plants but did include $^{14}C$
in the litter. The $^{14}C$-litter is obtained by multiplying the atmospheric value by the total carbon
in the litter:
$$Litter\ (^{14}C) = F^{14}_{atm} * Litter\ (C) \quad (4)$$
where $F^{14}_{atm}$ is the $F^{14}C$ of atmosphere at the time of leaf growth (figure 2).
Thus, from the litter, all processes defined in section 2.1 that apply to total soil carbon are also
represented for $^{14}C$.
We also take into account the radioactive decay of $^{14}C$. For that, we calculate the amount of
$^{14}C$ as follow:
$$^{14}C = {}^{14}C - K_{decrease} * {}^{14}C \quad (5)$$
Where $k_{decrease}$ is the radioactive decay constant ( $= Ln2/5730$) (Godwin, 1962)
The $F^{14}C$ of the soil is then calculated back for carbon, per pool:
$$F^{14}_{Pool,z} = \frac{^{14}C_{Pool,z}}{C_{Pool,z}} \quad (6)$$
with *pool* representing the active, slow or passive pool.
Finally, we calculate a mean $F^{14}C$ value per soil layer, according to the depth:
$$F^{14}_{Mean,z} = \frac{F^{14}_{active,z} * {}^{14}C_{active,z} + F^{14}_{slow,z} * {}^{14}C_{slow,z} + F^{14}_{passive,z} * {}^{14}C_{passive,z}}{^{14}C_{active,z} + {}^{14}C_{slow,z} + {}^{14}C_{passive,z}} \quad (7)$$

**2.3 Site descriptions**
**2.3.1 French sites**

Two Luvisol (WRB, 2006) profiles located in the northern France were selected: the Feucherolles and Mons sites. In Mons (49.87°N, 3.03°E), Luvisol, the soils sit under grassland, and are developed from several meters of loess and therefore well drained. The mean annual air temperature is 11°C and the annual precipitation is about 680 mm (Keyvanshokouhi et al., 2016). In Feucherolles (48.9°N, 1.97°E), the soil sits under oak forest and clay and gritstone deposits are found at approximately 1.5 m depth. The mean annual air temperature is 11.2°C and the annual precipitation is about 660 mm (Keyvanshokouhi et al., 2016). Both soils are neutral to slightly acidic and are characterized by the presence of a clay accumulation Bt horizon with clay content reaching 30 % for Feucherolles and 27 % for Mons, while the upper horizons are poorer in clay (17 % for Feucherolles and 20% for Mons).

The $^{14}$C data from the soils of both sites were obtained after chemical treatment done at Laboratoire des Sciences du Climat et de l'Environnement (LSCE) using a protocol adapted to achieve carbonate leaching without any loss of organic carbon; $^{4}$C activity was measured by AMS at the French Laboratoire de mesure du $^{14}$C (LMC14) facility (Cottereau et al., 2007). Details on measurements and sampling can be found in Jagercikova et al., (2017)

**2.3.2 Congo site**

The studied site is located in Kissoko (4.35°S, 11.75°E). It belongs to the SOERE F-ORE-T (Site de l'ObservatoirE de Recherche en Environnement sur le Fonctionnement des écosystèmes fOREsTiers) field observation sites of Pointe Noire, Republic of Congo. The mean annual air temperature is about 25°C with low seasonal variation (± 5°C), and average annual precipitation of 1400mm, and a dry season between June and September. The deep acidic sandy soil is a ferralic Arenosol (WRB, 2006). The soil is characterized by a sand content larger than 90% (Laclau et al., 2000). A soil profile was taken under native savanna vegetation dominated by C4 plants (Epron et al., 2009). The soil was sampled in May 2014 at different depths: 0-5cm, 5-10cm, 10-15cm, 15-20cm, 20-30cm, 30-40cm, 40-50cm, 50-60cm, 60-80cm, 80-100cm, 100-120cm. All samples were crushed and air-dried. Once in the laboratory, they were homogenized, crushed, randomly subsampled and sieved at 200µm. Then $^{14}$C measurements were made the same way as the two French sites, using the LSCE chemical treatment and the French LMC14 facility following recommendations by Cottereau et al., (2007).

**2.3.3 Argentina site**

The Province of Misiones is located in northeastern Argentina. The climate is subtropical humid without a dry season, an annual mean temperature of 20°C and 1850mm of mean annual rainfall (Morrás et al., 2009). The profile used in this study is located in the southern part of Misiones (27°S, 55°W). Native vegetation is a forest dominated by C3 plants. The soil selected is an Acrisol (WRB, 2006). It's a red clay soil, strongly to very strongly acid with a clay content varying from 40% at the surface to 60% at 1m depth. $^{14}$C measurements were made using a new Compact Radiocarbon System called *ECHo*MICADAS (Environment, Climate, Human, Mini Carbon Dating System) (Tisnérat-Laborde et al., 2015). Details on measurements and sampling can be found in Tifafi et al., *In prep*. Briefly, the soil was sampled in May 2015 at different depths: 0-5cm, 5-10cm, 10-15cm, 15-20cm, 20-30cm, 30-40cm, 40-50cm, 50-60cm, 60-80cm, 80-100cm. All samples were crushed and air-dried. Once

in the laboratory, they were homogenized, crushed, randomly subsampled and sieved at
200μm. Then [14]C measurements were made using a new Compact Radiocarbon System called
*ECHo*MICADAS (Environment, Climate, Human, Mini Carbon Dating System) following the
recommendations of Tisnérat-Laborde et al., (2015).
For the four sites, the SOC (kg m[-3]), for each depth $z$, was calculated using carbon content and
bulk density data using the following equation:
$$SOC_z = OCC_z * BD_z \qquad (8)$$
Where *OCC* (wt/wt) is the carbon content and BD (kg m[-3]) is the bulk density.

## 2.4 Different model tests

After the implementation of radiocarbon in the model, different tests were carried out (Table
2). Here we represent the outputs provided by three simulations:
i- Simulation using the initial version ORCHIDEE-SOM-[14]C (labelled "Control" in
figures and tables) in which no changes were made. The diffusion was kept constant
throughout the profile (D = 1.10[-4] m[2] year[-1]) and the other parameters are those of the
detailed version in Camino-Serrano et al., (2017).
ii- Simulation using the initial version ORCHIDEE-SOM-[14]C in which we modified
some parameters following He et al. (2016) ("He et al., (2016) parameterization" in
figures and tables). In brief, the authors used [14]C data from 157 globally distributed
soil profiles sampled to 1-meter depth to evaluate CMIP5 models. Their results show
that ESMs underestimated the mean age of soil carbon by a factor of more than six and
overestimated the carbon sequestration potential of soils by a factor of nearly two. So,
the suggestion (that we apply in this simulation) for the IPSL model was to multiply
the turnover time of the passive pool by 14 and the flux from slow pool to passive pool
by 0.07 (Table 2). The diffusion was kept constant throughout the profile (D = 1.10[-4]
m[2] year[-1]) but the turnover time of the passive pool increased from 462 years to 6468
years and the flux from the slow pool to the passive pool decreased from 0.07 to
268 0.0049.

iii- Simulation using the initial version ORCHIDEE-SOM-[14]C in which we assume that
the diffusion varies as a function of the depth ("Depth-varying diffusion constant" in
figures and tables) according to the equation below:
$$D(z) = 5.42.10^{-4}e^{(-0.04z)} \qquad (9)$$
Where $D$ is the diffusion (m[2] year[-1]) at a specific depth and $z$ is the depth. This equation of
diffusion varying as a function of depth following Jagercikova et al. ( 2014) and assumes that
bioturbation is higher in the top soil than in deep soil.

## 2.5 Model simulations

In order to reach a steady state of the soil module, we ran the model over 12700 years
(spinup). The state at the last time step of this spinup was used as the initial state for the
simulations. For this, the CRUNCEP meteorological data for the period 1901-1910 were used.
This has been applied for Misiones, Feucherolles and Mons. However, for Kissoko, a first
spinup similar to the other sites was carried out but a second one (over approximately 4200

years) was also done after the end of the first to take into account the change of the land cover from a tropical forest to a C4 savanna at this site (Schwartz et al., 1992). The atmospheric $CO_2$ concentration has been set at 296 ppm (year 1901, (Keeling and Whorf, 2006)) for the spinups and the $F^{14}C$ has been set to one corresponding to pre-industrial values. For each site, specific pH, clay content and bulk density values were used (Table 1). It should be noted that for these last data, only one value (the mean value on the profile) is provided as input for the model.

The simulations were outputted at a yearly time step, from 1900 to 2011. A yearly atmospheric $CO_2$ concentration value (Keeling and Whorf, 2006) is read for the sites. The same specific pH, clay content and bulk density values were used (Table 1).

Figure 2 shows the evolution of the $F^{14}C$ values in the atmosphere used in our model for Argentina, Congo and France (Figure 5 from Hua et al. (2013)). The values provided are classified into five zones, three in the Northern Hemisphere (NH) and two in the Southern Hemisphere (SH), corresponding to different levels of $^{14}C$. For France, the values correspond to the NH zone 2, for the Congo to the SH zone 3 and finally for Argentina to the SH zone 1-2. Thus, for our simulations, a yearly value is read for each site.

An $F^{14}C$ value of 1.8 represents a doubling of the amount of $^{14}C$ in atmospheric $CO_2$. In figure 2, it can be noted that the values recorded in France (northern hemisphere) are higher than those in the Congo and Argentina (southern hemisphere). This is due to the preponderance of atmospheric tests in the northern hemisphere and the time required to mix air across the equator.

**2.6 Statistical analysis**

Simulating carbon processes in soil requires comparison between the model outputs and the measurements to test the model accuracy and possibly implement further improvement. Statistical analysis based on the statistics of deviation were done to evaluate the model–measurement discrepancy according to Kobayashi and Salam (2000) (where a detailed description of the method is provided). Here, we only reproduce the different equations used. $x$ refers to the model outputs and $y$ to the measurements, while $i$ refers to soil depth. The intervals of soil depth of the model outputs and the measurements were homogenized by linearly interpolating the data to common depth intervals defined for each site. The simulations and data were then compared for each depth interval.

$$RMSD = \sqrt{\frac{1}{n}\sum_{i=1}^{n}(x_i - y_i)^2} \tag{10}$$

RMSD is the Root Mean Squared Deviation, which represents the mean distance between simulation and measurement.

$$MSD = \frac{1}{n}\sum_{i=1}^{n}(x_i - y_i)^2 = (\bar{x} - \bar{y})^2 + \frac{1}{n}\sum_{i=1}^{n}[(x_i - \bar{x}) - (y_i - \bar{y})]^2 \tag{11}$$

MSD, the Mean Squared Deviation, is the square of RMSD. The lower the value of MSD, the closer the simulation results are to the measurements.

$$SB = (\bar{x} - \bar{y})^2 \tag{12}$$

Where are the means of $x_i$ (model outputs) and $y_i$ (measurements) respectively.

SB is a part of the MSD (Eq.14) and represents the bias of the simulation from the
measurement.
$$SD_s = \sqrt{\frac{1}{n}\sum_{i=1}^{n}(x_i - \bar{x})^2} \qquad (13)$$
$SD_s$ is the Standard Deviation of the simulation.
$$SD_m = \sqrt{\frac{1}{n}\sum_{i=1}^{n}(y_i - \bar{y})^2} \qquad (14)$$
$SD_m$ is the Standard Deviation of the measurements.
$$r = \frac{\frac{1}{n}\sum_{i=1}^{n}(x_i - \bar{x}) - (y_i - \bar{y})}{SD_m \, SD_s} \qquad (15)$$
$r$ is the correlation coefficient between the simulation and measurements.
$$SDSD = (SD_s - SD_m)^2 \qquad (16)$$
SDSD  is the difference in the magnitude of fluctuation between the simulation and
measurements.
$$LCS = 2SD_s \, SD_m (1 - r) \qquad (17)$$
LSC represents the lack of positive correlation weighted by the standard deviations.
The MSD can be therefore be rewritten as:
$MSD = SB + SDSD + LCS \qquad (18)$
For the different simulations, the MSD and its components were calculated according to the
total soil carbon and to the $F^{14}C$.

### 3 Model results and evaluation

### 3.1 Outputs from simulation using the initial version of the model ORCHIDEE-SOM-$^{14}$C (Control)

### 3.1.1 Simulated total soil carbon

Results from the initial version of ORCHIDEE-SOM-$^{14}$C show that in all the studied sites, the
model succeeds in reproducing the trend of the total carbon profiles, with more carbon at the
surface which then decreases according to the depth (Figure 3). Moreover, total soil carbon
stock simulated down to 2m depth is in accordance with data in the case of Misiones and
Feucherolles where the major difference mainly lies on the surface. This results in correlation
coefficients of 0.44 and 0.2 respectively (Table 3). For the sites of Kissoko and Mons, an
over-estimation of the total soil carbon is found to a depth of 50cm for Kissoko  and up to a
depth of 120cm  for Mons. Correlation coefficients are 0.14 and 0.49 for Kissoko and Mons
respectively (Table 3).
Metrics presented in Figure 4, showed that this version (ORCHIDEE-SOM-$^{14}$C) represents
relatively well the observation from Feucherolles (MSD = 206 kg C m$^{-6}$), whereas the other
are highly overestimated (Kissoko, MSD = 1343 kg C m$^{-6}$; Misiones MSD = 2180 kg C m$^{-6}$;
Mons MSD = 3355 kg C m$^{-6}$). By detailing the different components of the MSD (Figure 4),

we note that for Mons and Kissoko, standard bias (SB) is the major component of the MSD with contributing 70% and 60% respectively. This reflects that the average of total soil carbon over the soil profile simulated by the model is primarily the origin of the deviation of the model outputs from data. The mean total soil carbon estimated by the model (Table 3) is almost three times higher than the mean total carbon measured for Mons (2.37 kg C m$^{-2}$ against 0.8 kg C m$^{-2}$ respectively) and it is more than five times that measured for Kissoko (2.44 kg C m$^{-2}$ against 0.42 kg C m$^{-2}$ respectively). For Mons a net primary production (NPP) of 6.7 t ha$^{-1}$ yr$^{-1}$ was estimated by the technical institute for pasture in this region of France based on the annual yields, whereas the model predicts a NPP of 7.5 t ha$^{-1}$ yr$^{-1}$. The large overestimation of the SOC stocks may therefore be due to an overestimation of the NPP. This significant gap recorded in the case of the Kissoko site, where the measured SOC is very low, is probably due to an overestimation of decay rates by ORCHIDEE in sandy soils. The correlation coefficient for Mons is relatively high compared to other site (Table 3) whereas Fig. 3 shows that the model performance was not very good for this site. This is mainly due to a large SB whereas other MSD components were rather low.

However, the main components of MSD for Feucherolles and Misiones are both SB (46% and 56% for Feucherolles and Misiones, respectively) and also LCS (53 and 31% for Feucherolles and Misiones, respectively). This means that for these two sites, the deviation between model outputs and measurements is mainly due to a variation of carbon stock estimation throughout the profile. The mean total soil carbon estimated in these both cases (Table 3) is only slightly higher than those measured (2.03 kg C m$^{-2}$ estimated against 2.14 kg C m$^{-2}$ measured for Misiones and 0.7 kg C m$^{-2}$ estimated against 0.68 kg C m$^{-2}$ measured for Feucherolles).

The vertical profiles of the SOC stock were fairly represented by the model. The overestimation, especially at the top, suggests that the distribution of the litter following the root profile and / or the vertical transport of SOC by diffusion are not correctly described in the model.

### 3.1.2 Simulated F$^{14}$C

Regarding the $^{14}$C activity, bulk F$^{14}$C profiles show a classical pattern with higher $^{14}$C activity on the top, slightly influenced by the peak bomb enriched years. Subsequently profiles show decreasing $^{14}$C activity with depth (Figure 5).

The estimated profiles (Model-Control) follow the same trend with a decrease from the surface to the depth. However, there is a significant difference between the estimated values and those measured throughout the profile. The statistical analyzes (Figure 6) provide MSD values: 0.02 for Mons and Misiones, 0.03 for Kissoko and 0.09 for Feucherolles. The major component of the MSD in the four sites is the LCS, with a proportion reaching 90% for Mons, 80% for Misiones and 70% for Congo, but only 55% for Feucherolles. The high proportions of LCS suggest that the model fails to reproduce the shape of the profile. The lower values estimated by the models reflect a more modern carbon age than in reality. This can be explained, first, by the fact that the root profile puts too much fresh organic carbon in deep

soil. Afterwards, in ORCHIDEE, root profile is assumed to follow an exponential function without modulation due to environmental conditions.

SB's contribution to the MSD does not exceed 7% for Misiones, Kissoko and Mons but reaches about 40% for Feucherolles. This reflects that the mean value of the $F^{14}C$ estimated by the model and that obtained after the measurements are not very different, except for Feucherolles site (Table 4). Indeed, the average value estimated for Misiones is 0.920, very close to that measured at 0.930, 0.995 for Kissoko against 0.985 measured and 0.860 for Mons against 0.815 measured. Yet, the difference is greater for the Feucherolles site, the estimated value being 0.915 while the measurement is 0.725. This difference might be caused by the low $F^{14}C$ value measured at 150cm (0.257), that the model is not able to capture. This suggests that modeled deep soil carbon is much younger than the observed total soil carbon, probably because ORCHIDEE-SOM simulates a relatively small proportion of passive pool in the lower soil horizons (Figure 7), while an increasing proportion of passive carbon with soil depth could be expected.

In brief, SOC stocks are generally overestimated and soil carbon age in deep soils (as shown by the $F^{14}C$) is underestimated, suggesting that the turnover rate of the passive pool is subject to improvements in ORCHIDEE-SOM.

### 3.2 Outputs from simulation using the initial version of the model ORCHIDEE-SOM-$^{14}C$ including He's suggestion (He et al., (2016) parameterization)

### 3.2.1 Simulated total soil carbon

Figure 3 shows profiles output after He et al (2016)'s suggestion was implemented into ORCHIDEE-SOM-$^{14}C$ (green dotted curves). Resulting profiles follow the same trend than observations but in this case (''He et al., (2016) parameterization''), the overestimation is very high across the whole profile. This is further confirmed by the metrics analysis (Figure 4). MSD values markedly increased, resulting in an even higher variance. Obviously, the major component of MSD in all cases is the SB (varying from 80% to 87%) reflecting an even more marked overestimation of the mean total carbon estimates: 7.38 kg C m$^{-2}$ against 2.14 kg C m$^{-2}$ for Misiones, 2.44 kg C m$^{-2}$ against 0.42 kg C m$^{-2}$ for Kissoko, 2.33 kg C m$^{-2}$ against 0.66 kg C m$^{-2}$ for Feucherolles and 9.99 kg C m$^{-2}$ against 0.8 kg C m$^{-2}$ for Mons.

### 3.2.2 Simulated $F^{14}C$

He et al., (2016) parameterization outputs (Figure 5, green dotted curves) for $F^{14}C$ are once again even further away from observations and MSDs (Figure 6) are much higher, except for Feucherolles. The MSD components for the Feucherolles site show that the LCS increases from 0.05 to 0.06 whereas the SB decreases from 0.04 to 0.03, again reflecting a variation of the profile more than a difference from the means.

Improvement of the model-measurement fit for the $F^{14}C$ at 150 cm in Feucherolles confirms that the deep soil carbon simulated by the control version of ORCHIDEE-SOM-$^{14}C$ was excessively young, since the longer residence time of the passive pool reported by He et al. (2016) resulted in a higher proportion of passive pool across the soil profile (Figure 7), thus improving deep soil carbon age. Nevertheless, this test only improves the simulation of deep

soil carbon in Feucherolles. On the contrary, this increase in carbon residence time increases
model deviation from observations for all the other cases (Figure 5 and 6).
Indeed, taking the priming effect into account in this new version of ORCHIDEE has
contributed to a 50% of decrease in carbon storage over the historical period. He et al.,
(2016)'s correction was also aimed at reducing this storage and is of the same order of
magnitude as the priming effect. Thus, applying He's correction to this version of the model,
which takes into account the priming effect, contributes to a double correction for the same
target, which then generates this important difference between model outputs and
measurements. Moreover, the work of He et al. (2016) is done under the standard
parameterization of ORCHIDEE based on Century, while ORCHIDEE-SOM was re-
parameterized after adding several different processes, the priming effect among them
(Camino-Serrano et al., 2017), which makes it difficult to compare results from between the
two studies.

**3.3 Outputs from simulation using the initial version of the model ORCHIDEE-SOM-**
**$^{14}$C with diffusion varying according to the depth (Depth-varying diffusion constant)**
**3.3.1 Simulated total soil carbon**
Fick's law of diffusion is classically used in models to represent bioturbation assuming that
soil fauna activity may be represented following the Fick's law of diffusion (Elzein and
Balesdent, 1995; Guenet et al., 2013; Koven et al., 2013; O'Brien and Stout, 1978; Wynn et
al., 2005). Using a fixed diffusion constant ($D$ in eq. 2) implicitly suggests that soil fauna
activity is uniform over the entire soil profile. This is generally the case of several models of
diffusion, in particular at the level of an ecosystem (Bruun et al., 2007; Guimberteau et al.,
2018; O'Brien and Stout, 1978). However soil faunal activity vary naturally with depth and
the diffusion constant should therefore be depth-dependent (Jagercikova et al., 2014).
With Depth-varying diffusion constant, the carbon profiles (orange dashed curves) was
improved compared to the initial outputs (Control). The overestimation at the surface
decreases at the four sites (Figure 3). In particular, the Misiones outputs fit very well the
observed profiles. This is confirmed with lower MSDs for the four sites for this version
compared to Control (Figure 4).
The total SOC stocks simulated according to this third simulation are closer to the measured
values and describing the vertical transport of SOC through diffusion varying according to the
depth improves significantly the model outputs.
**3.3.2 Simulated F$^{14}$C**
Regarding the F$^{14}$C outputs, the simulations using the initial version ORCHIDEE-SOM-$^{14}$C in
which we assume that the diffusion varies as a function of the depth (Depth-varying diffusion
constant) results in an improvement of the F$^{14}$C profiles (orange dashes curves), in particular
for the sites Misiones, Mons and Kissoko (Figure 5). Statistical analyzes prove it with
significantly lower MSDs. In addition, the proportion of LCS is 98%, 92% and 88% for
Mons, Misiones and Kissoko, respectively, highlighting an estimated average very close to
the measurements with a clear disparity, less marked than with the first two simulations,
throughout the profile (Figure 6). Overall, the simulated F$^{14}$C to 2 m of depth according to

this third simulation are in a better agreement with the measured values, and thus incorporating diffusion that varies with depth significantly improves the model outputs.

Using a diffusion coefficient that varies as a function of the depth seems to correct the overestimation of the surface total soil carbon by increasing the proportion of labile soil carbon pools in the first soil layers.

When we sum the total soil carbon at each soil layer and look at the relative proportion of each of the soil carbon pools (Figure 7), we note that it is mainly the distribution of the litter according to the depth which varies. In fact, the structural litter proportion is multiplied by about 2 in all four cases, and this proportion remains relatively constant across the profile. This increase in litter proportion has also resulted in a decrease in the passive pool, more pronounced at the surface but also important at depth (except for Feucherolles where the decrease is only marked at the bottom). It suggests that the vertical carbon distribution, which is largely modified by the diffusion coefficient, greatly impacts the SOC and $^{14}$C profiles, which is in line with Dwivedi et al. (2017) who found that the vertical carbon input profiles were important controls over the $^{14}$C depth distribution.

In this study, the vertical transport of SOC and litter through diffusion has been improved by varying diffusion according to the depth. Further model development should explore the impact of the other processes defining the soil carbon pools vertical distribution and especially the distribution of the litter according to the root profile.

Overall, by using radiocarbon ($^{14}$C) measurements we have been able to diagnose internal model biases (underestimation of deep soil carbon age) and to propose further model improvements (depth-dependent diffusion). Therefore, the use of radiocarbon ($^{14}$C) tracers in global models emerges as a promising tool to constrain not only SOC turnover times in the long-term (He et al., 2016), but also internal SOC processes and fluxes that have no direct comparison with field measurements. Nevertheless, the model evaluation performed here on only four sites should be considered as proof of concept and more in depth evaluation are needed, in particular using a large $^{14}$C database available at global scale (Balesdent et al., 2018; Mathieu et al., 2015). Indeed, the F$^{14}$C is largely controlled by pedo-climatic conditions such as clay content, climate and mineralogy (Mathieu et al., 2015) and the range of situations we covered here is relatively limited.

## 4 conclusion

ORCHIDEE-SOM-$^{14}$C, is one of the first land surface models that incorporates the $^{14}$C dynamics in the soil (Koven et al., 2013). Its starting point is ORCHIDEE-SOM, a recently developed soil model. We evaluated the new model ORCHIDEE-SOM-$^{14}$C for four sites in different biomes. The model almost managed to reproduce the soil organic carbon stocks and the $^{14}$C content along the vertical profiles at all four sites. However, an overestimation of the total carbon stock throughout the profile was noted, with the greatest deviationat the surface. By using radiocarbon ($^{14}$C) measurements, we have been able to diagnose internal model biases (underestimation of deep soil carbon age) and to propose further model improvements (depth-dependent diffusion). These results demonstrate the importance of depth-dependent

diffusion to improving model outputs with regards to observations. This suggests that, from now on, model improvements should mainly focus on a depth dependent parameterization. We limited our work here to depth-varying diffusion, but other parameters are also depth dependent and should be represented as such in the next version of the model. For instance, belowground litter production in the model is simply represented by an exponential law without any representation of the effect of resource distribution on root profile (e.g. water or nutrients). This is a complex task in a land surface model running at large scale with a classical resolution of 0.5°, but the soil modules of land surface models are quite sensitive to the NPP (Camino-Serrano et al., 2018; Todd-Brown et al., 2013) and a better constraint on the profile of the below ground litter production would likely improve the model performance. Furthermore, here we used only one averaged value over the soil profile for soil boundary conditions (texture, pH, bulk density) but those variables are known to impact the $F^{14}C$ (Mathieu et al., 2015) and change with depth (Barré et al., 2009) and depth-varying boundary conditions may also help to improve the model. Finally, the next step will deal with the comparison of model outputs to data at larger scales to be able to run the new version ORCHIDEE-SOM-$^{14}C$ at both regional and global scales.

**Code availability**

The version of the code is freely available here:

http://forge.ipsl.jussieu.fr/orchidee/wiki/GroupActivities/CodeAvalaibilityPublication/ORCHIDEE_gmd-2018-14C

**Acknowledgement**

This study, part of the MT's PhD, financed by the University of Versailles Saint Quentin, is within the scope of the ANR-14-CE01-0004 DeDyCAS project. Marta Camino-Serrano acknowledges funding from the European Research Council Synergy grant ERC- 2013-SyG-610028 IMBALANCE-P. Part of the data were acquired in the frame of the AGRIPED project (ANR 2010 BLAN 605). We thank Matthew McGrath for his valuable comments on the manuscript.

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

**Table 1.** General description of the studied sites. The mean bulk density, pH and clay fraction values calculated from the different soil layers depths available from the data were used as input for each site. For the Mons and Feucherolles sites, min and max values of pH and clay fraction are provided between brackets.

| Site name | Feucherolles | Mons | Kissoko | Misiones |
|---|---|---|---|---|
| Sampling Date | April 2011 | March 2011 | May 2014 | May 2015 |
| Location | France | France | Congo | Argentina |
| Coordinates | 48.90°N, 1.97°E | 49.87°N, 3.03°E | 4.35°S, 11.75°E | 27.65°S, 55.42°W |
| Elevation (m) | 120 | 88 | 100 | NA |
| Mean Annual Rainfall (mm) | 660 | 680 | 1400 | 1850 |
| Mean Annual Temperate (°C) | 11.2 | 11 | 25 | 20 |
| Soil Type (WRB) | Luvisol | Luvisol | Arenosol | Acrisol |
| Land Use | Temperate broad-leaved summergreen forest | Grassland | Native savanna | Tropical broad-leaved evergreen forest |
| Mean Bulk Density (g cm$^{-3}$) | 1.34 | 1.4 | 1.48 | 1.15 |
| Mean pH | 5.9 (5.12-8.55) | 6.9 (6.70-7.56) | 5.2 | 5.2 |
| Mean Clay Fraction (%) | 20 % (13-30 %) | 23 % (19-27 %) | 5 % | 58 % |

**Table 2.** The main differences between the three simulations

| | Flux from slow pool to passive pool | Turnover time of the passive pool (year) | Diffusion (m$^2$ year$^{-1}$) |
|---|---|---|---|
| Control | 0.07 | 462 | D(z) = 1.10$^{-4}$ |
| He et al., (2016) parameterization | 0.0049 | 6468 | D(z) =1.10$^{-4}$ |
| Depth-varying diffusion constant | 0.07 | 462 | $D(z) = 5.42.10^{-4}e^{(-0.04z)}$ |

**Table 3.** The correlation coefficient (r) between model outputs and measurements for carbon stock (kg C m$^{-2}$) over the soil profile, for the four sites. The results of the initial version of the model ORCHIDEE-SOM-$^{14}$C (Control) as well as those from the version including the modification according to (He et al., 2016) (He et al., (2016) parameterization) and diffusion varying according to the depth (Depth-varying diffusion constant) are provided.

| | | r | Mean total soil carbon (kg C m$^{-2}$) Model | Mean total soil carbon (kg C m$^{-2}$) Measurements |
|---|---|---|---|---|
| **Misiones** | Control | 0.44 | 2.03 | |
| | He et al., (2016) parameterization | 0.69 | 7.38 | 2.14±0.30 |
| | Depth-varying diffusion constant | 0.46 | 2.23 | |
| **Kissoko** | Control | 0.14 | 0.76 | |
| | He et al., (2016) parameterization | 0.55 | 2.44 | 0.42±0.38 |
| | Depth-varying diffusion constant | 0.13 | 0.88 | |
| **Feucherolles** | Control | 0.20 | 0.70 | |
| | He et al., (2016) parameterization | 0.11 | 2.33 | 0.66±0.08 |
| | Depth-varying diffusion constant | 0.22 | 0.77 | |
| **Mons** | Control | 0.49 | 2.37 | |
| | He et al., (2016) parameterization | -0.14 | 9.99 | 0.8±0.10 |
| | Depth-varying diffusion constant | 0.48 | 2.42 | |

**Table 4.** The correlation coefficient (r) between model outputs and measurements and the mean values (provided by the model and the measurements) over the profile according to F$^{14}$C for the four sites. The results of the initial version of the model ORCHIDEE-SOM-$^{14}$C (Control) as well as those from the version including the modification according to (He et al., 2016) (He et al., (2016) parameterization) and diffusion varying according to the depth (Depth-varying diffusion constant) are provided.

| | | r | Mean Model | Mean Measurements |
|---|---|---|---|---|
| **Misiones** | Control | 0.55 | 0.920 | |
| | He et al., (2016) parameterization | 0.50 | 0.560 | 0.930±0.009 |
| | Depth-varying diffusion constant | 0.60 | 0.900 | |
| **Kissoko** | Control | 0.40 | 0.995 | |
| | He et al., (2016) parameterization | 0.30 | 0.620 | 0.985±0.004 |
| | Depth-varying diffusion constant | 0.55 | 0.995 | |
| **Feucherolles** | Control | 0.55 | 0.915 | |
| | He et al., (2016) parameterization | 0.55 | 0.550 | 0.725±0.005 |
| | Depth-varying diffusion constant | 0.60 | 0.890 | |
| **Mons** | Control | 0.75 | 0.860 | |
| | He et al., (2016) parameterization | 0.70 | 0.510 | 0.815±0.005 |
| | Depth-varying diffusion constant | 0.80 | 0.835 | |

**Table 5.** $F^{14}C$ profile obtained for each site.

| Sites | Soil depth (cm) | $F^{14}C$ |
|---|---|---|
| Misiones | 0-5 | 1.08 |
| | 5-10 | 1.04 |
| | 10-15 | 1.05 |
| | 15-20 | 0.99 |
| | 20-30 | 0.99 |
| | 30-40 | 0.87 |
| | 40-50 | 0.91 |
| | 50-60 | 0.76 |
| | 60-80 | 0.79 |
| | 80-100 | 0.79 |
| Kissoko | 0-5 | 1.06 |
| | 5-10 | 1.07 |
| | 10-15 | 1.07 |
| | 15-20 | 1.08 |
| | 20-30 | 1.05 |
| | 30-40 | 1.04 |
| | 40-50 | 1.02 |
| | 50-60 | 0.97 |
| | 60-80 | 0.90 |
| | 80-100 | 0.81 |
| | 100-120 | 0.72 |
| Feucherolles | 0-2 | 1.08 |
| | 16-18 | 1.05 |
| | 40-45 | 0.92 |
| | 75-85 | 0.69 |
| | 105-115 | 0.54 |
| | 125-135 | 0.53 |
| | 147-157 | 0.26 |
| Mons | 0-2 | 1.02 |
| | 2-4 | 1.03 |
| | 18-20 | 1.03 |
| | 45-50 | 0.87 |
| | 60-65 | 0.71 |
| | 82-92 | 0.65 |
| | 102-112 | 0.64 |
| | 142-152 | 0.55 |


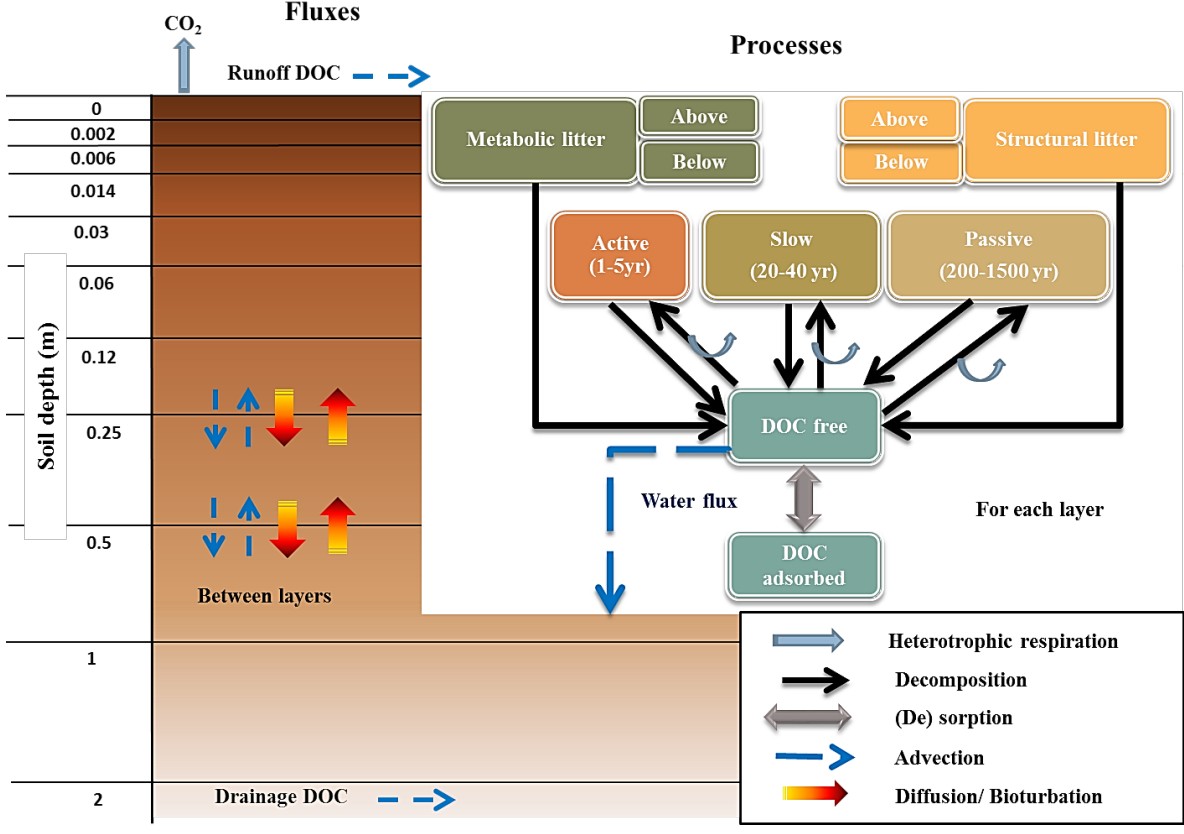

**Figure 1.** Overview of the different fluxes and processes in soil as presented in the version of
ORCHIDEE-SOM adapted from Camino-Serrano et al. (2017).

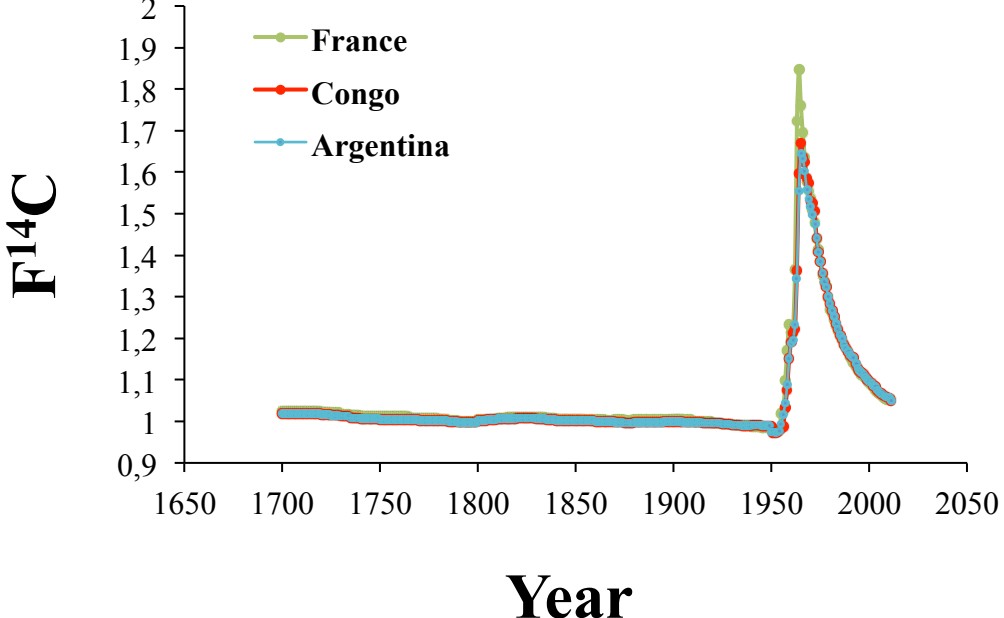

**Figure 2.** Evolution of the $F^{14}C$ of atmospheric $CO_2$ in Argentina, Congo and France (data from Hua et al. 2013).

933

934

**Figure 3.** Total soil carbon (kg C m$^{-3}$) according to the depth for the four sites. The results of
the initial version of the model ORCHIDEE-SOM-$^{14}$C (Control) as well as those from the
version including the modification according to  (He et al., 2016) (He et al., (2016)
parameterization) and diffusion varying according to the depth (Depth-varying diffusion
constant) are shown.






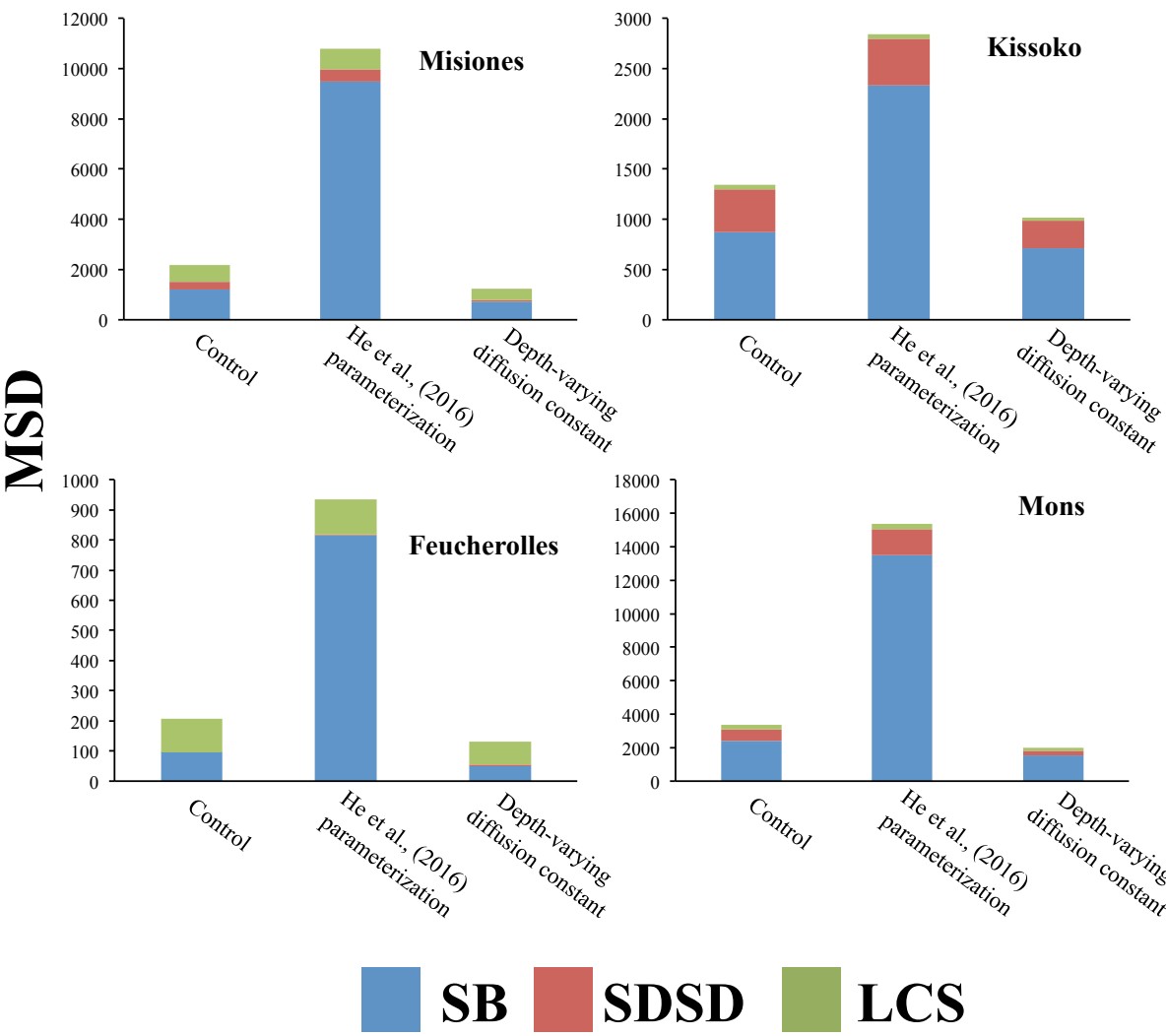


**Figure 4.** Mean Squared Deviation (MSD) and its components for total soil carbon (kg C m$^{-6}$): lack of correlation weighted by the standard deviation (LCS), squared difference between standard deviations (SDSD) and the squared bias (SB). For the four sites, the results of the initial version of the model ORCHIDEE-SOM-$^{14}$C (Control as well as those from the version including the modification according to (He et al., 2016) (He et al., (2016) parameterization) and diffusion varying according to the depth (Depth-varying diffusion constant), are shown.






# F$^{14}$C

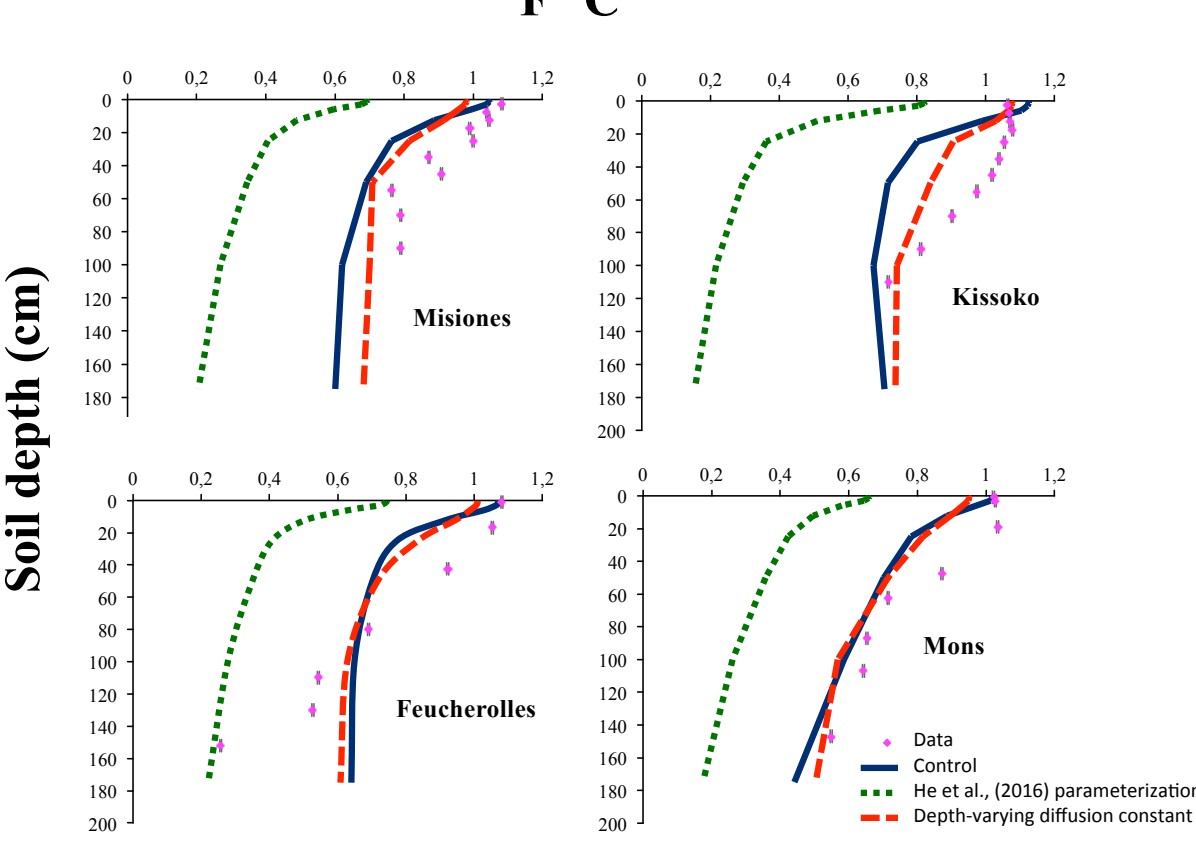

**Figure 5.** Modern fraction F$^{14}$C according to the depth, for the four sites. The results of the initial version of the model ORCHIDEE-SOM-$^{14}$C (Control) as well as those from the version including the modification according to He et al., (2016) (He et al., (2016) parameterization) and diffusion varying according to the depth (Depth-varying diffusion constant) are shown.

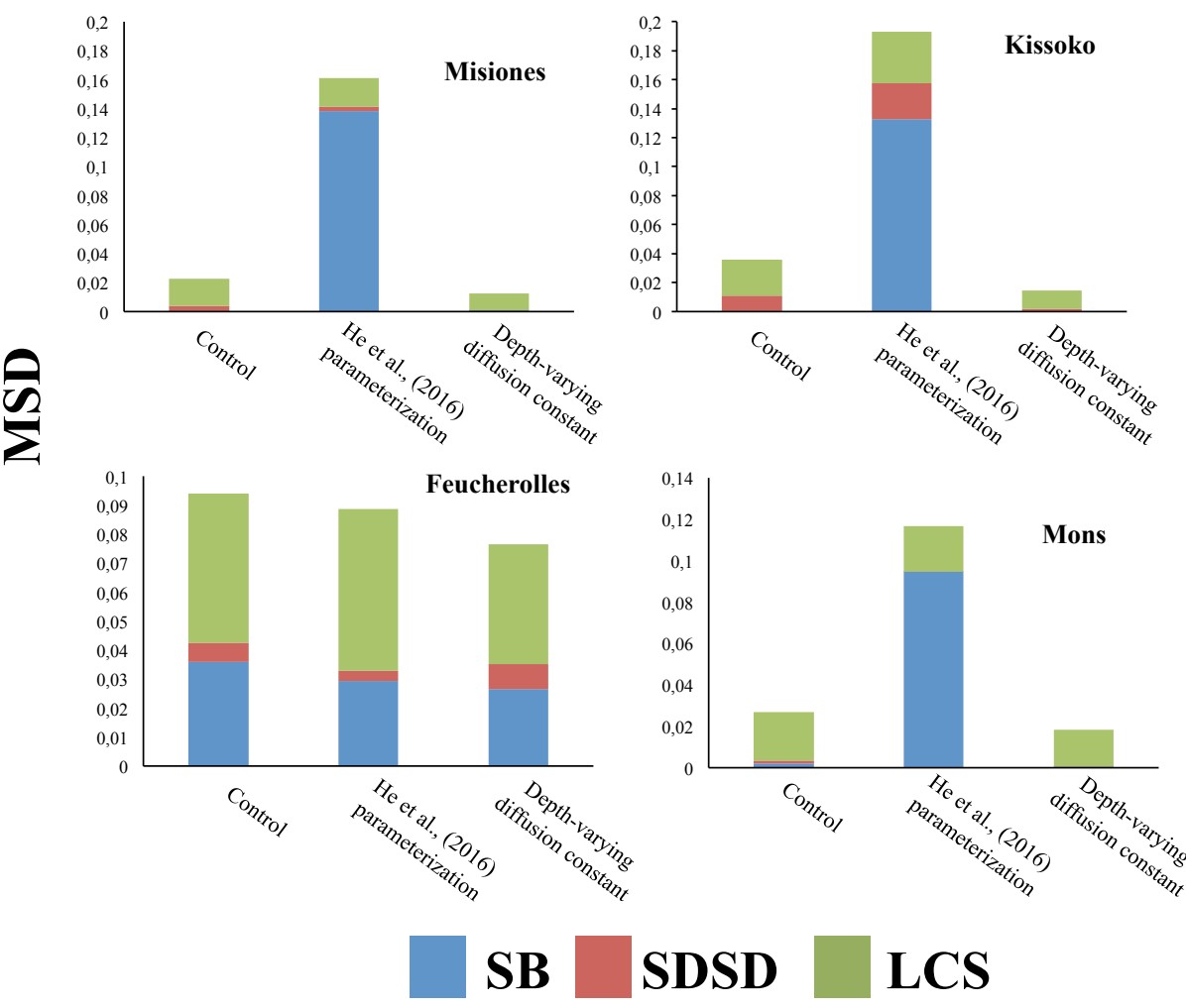

**Figure 6.** Mean Squared Deviation (MSD) and its components: lack of correlation weighted by the standard deviation (LCS), squared difference between standard deviations (SDSD) and the squared bias (SB) calculated for modern fraction $F^{14}C$. For the four sites, the results of the initial version of the model ORCHIDEE-SOM-$^{14}C$ (Control) as well as those from the version including the modification according to He et al., (2016) (He et al., (2016) parameterization) and diffusion varying according to the depth (Depth-varying diffusion constant) are shown.

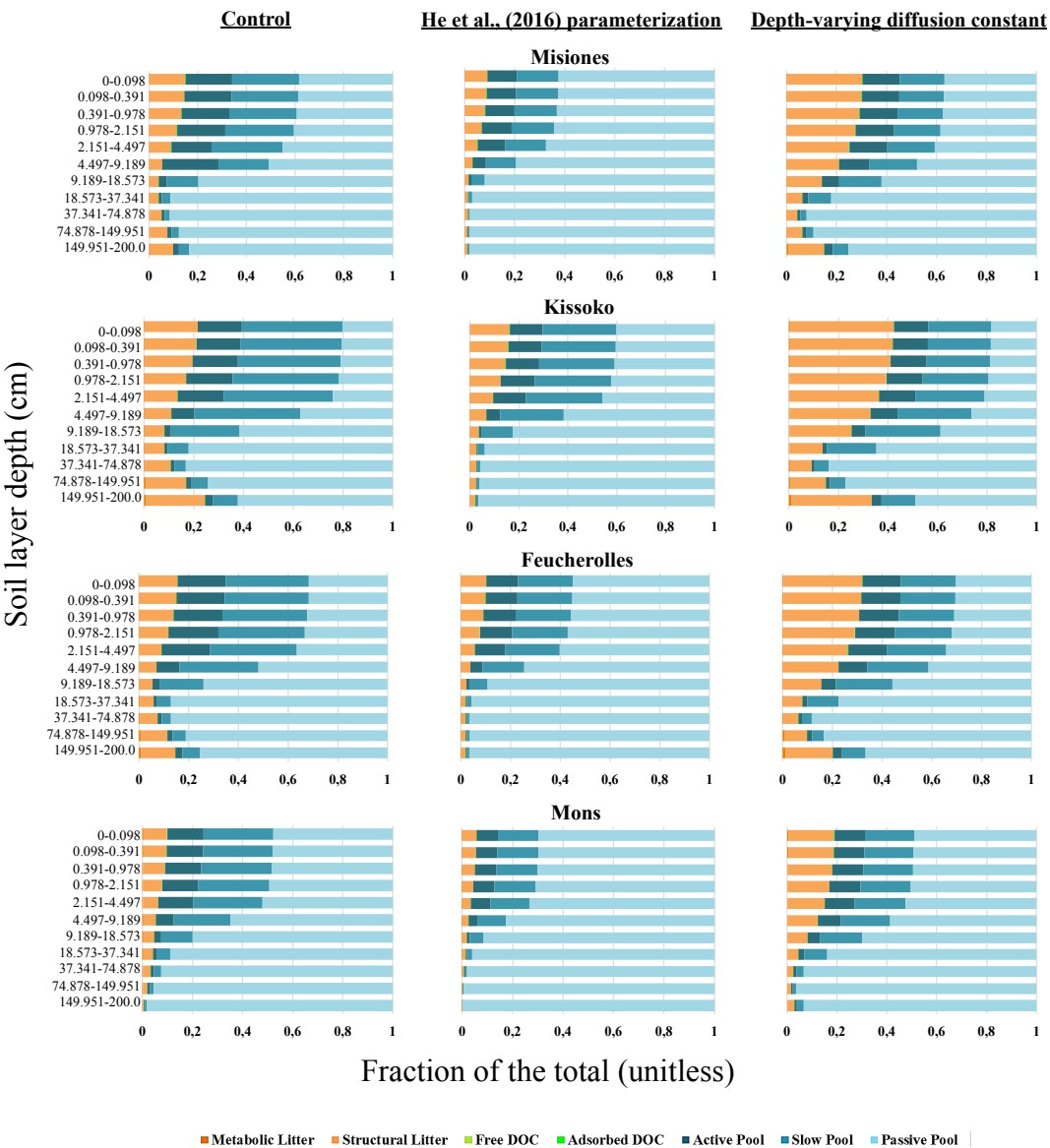

**Figure 7.** Relative proportion of each of the soil carbon pools summing the total soil carbon at each soil layer. The results of the initial version of the model ORCHIDEE-SOM-$^{14}$C (Control, left pattern) as well as those from the version including the modification according to (He et al., 2016) (He et al., (2016) parameterization, pattern in the middle) and diffusion varying according to the depth (Depth-varying diffusion constant, right pattern) are shown.