# Peer review of "The use of radiocarbon 14C to constrain carbon dynamics in the soil module of the land surface model ORCHIDEE (SVN r5165)"

_Geoscientific Model Development, 2018_

## Referee Comment (RC1) · Anonymous Referee #1 · 14 Jun 2018

Review of the manuscipt The use of radiocarbon 14C to constrain carbon dynamics in the soil module of the land surface model ORCHIDEE (SVN r5165)

Improvement of the soil modules in global carbon cycle models is a recurrent need, claimed by the scientifc community. The land surface model Orchidee is one of the important tools to analyse and predict future changes of the Earth's climate and biosphere. A recent study highlighted that current Earth system models predict a too young age for soil organic carbon. The present work introduces the radiocarbon isotope in the model to better constrain Orchidee. Based on the use of radiocarbon, the study furthermore improved the model itself, and model prediction, through a better

representation of carbon movement within the soil profile. The article is fully relevant, clearly written and illustrated, and worth publication in GMD.

But one point requires a significant change. Once this point fixed, the paper could be acceptable with only minor corrections.

Important point. (parameterization of the 'Model_Test_He') Line 233 authors state "... multiply by 14 the turnover rate and by 0.07 the flux...". And later Line 236 and in Table 2 : "decrease the flux from 0.07 to 0.049. Consistency would either decrease the flux from 0.07 to 0.0049, or multiply it by a factor 0.7. I suppose that the initial intention of authors was to multiply by 0.07 the flux, so that the steady state stock of passive would be kept similar (multiplied by 14 x 0.07 = 0.98), but with a F14 much lower. Here it seems from the results that the stock of the passive pool was multiplied by a factor almost 10 (less than 10 because of the duration of the spin-up), as expected by a factor x 0.7 for the flux. The over estimation of both carbon content and age is obviously expected with such a parameterization. In the present state, I further recomment not to use the name of a person in the surname of the model.

Finally, I recommend that the authors either (i) remove this model_test from the paper, which would then be accepted with minor revison, or (ii) recalculate using a flux to passive 0.0049 instead of 0.049. Option (ii) is preferred, but is not mandatory, since the other parts bring significant results; note that option (i) would not affect the summary nor the conclusion.

Minor and typographical points.

Table 1. Be clear in the legend on what was averaged. Do "over the profiles" means a calculated mean for (0- 2.0 m)" ?

Table 3 Data are in kg C m-3, which is a unit for local concentration, not for carbon stock. Is it: (a) kg C m-2, i.e., the carbon stock per unit area; or (b) the average concentration over the 0-2.0 m profile (then the Stock would be 2 times the mean

concentration value) ? Option (a) would be preferred.

Line 786 Legend fig.6: indicate the variable in object (= F14C).

Line 143. A brief statement of the formalism and parameterization of the priming would be welcome.

Line 179. In Eq(6), STRUC was excluded of total 14C. Was it ? In the Century model, STRUC usually accounts for 10- 20% of C in 0-20 cm layer, and is therefore non negligible (your figure 7). It is considered as retreived as material < 2mm (for a large part) and therefore often included in the "measured" total carbon. This exclusion may affect the comparison between observed and modelled values of F14.

Line 218. "OCC (wt/wt)" would be better than "OCC (wt %)"

Line 232. "turnover rate" is an unclear term (might be the reciprocal of turnover time). Here turnover time ?

Lines 314-326 and throughout: MSD values aren't in kg C m-3, but in kg2 C m-6 (variance not standard deviation); or use squareroot(MSC)

Line 325. Arenosols are not very specific and are broadly represented on the planet. Remove "for such specific conditions". Replace by " Probably due to an overestimation of decay rates by ORCHIDEE in sandy soils ?

Lines 367-401. See Major comment.

Do is the boundary condition at depth 2.0 for constant diffusion affect the base of the profils ?

Typography

Line 71. "this"

Line 158. verify in the final edition the greek symbol delta (not ok in my pdf)

Line 186. "1.5_m" (= separate the units throughout)

Line 227. "(2016) (=no square brackets)

Line 242. "et al. (2014)" (spaces)

Line 255. Point missing; also lines 326, 337 ... check.

Line 315: spaces before and after "=" (throughout the text)

Line 447 'processes'

---

## Referee Comment (RC2) · Anonymous Referee #2 · 18 Aug 2018

This paper presents ORCHIDEE-SOM-14C, a new version of the IPSL-Land Surface Model, and tests it against data from four different sites. It makes an important contribution by implementing the isotopic tracer 14C in the model. This is a valuable addition to the ORCHIDEE-SOM model, which simulates depth-resolved soil carbon dynamics from 0-2m below the surface. The authors also demonstrate how the new model can be used to constrain SOC turnover times and internal model processes. In particular, they implement two variations on the model ("Model_Test_He" and "Model_Test_Diffusion"). They follow the suggestions of He et al (2016) to slow turnover in the passive pool and reduce the flux from the slow to passive pool (pending comment by reviewer #1). They also implement a version of the model with depth-dependent bioturbation rates,

following Jagercikova et al (2014). Conceptually, this paper is a nice demonstration of how F14C data could be used for comparison against different model implementations. However, there are significant issues which should be addressed both with the implementation (see Reviewer #1 comments) and interpretation of the results (see below) prior to publication.

In its current form, this paper does not convincingly demonstrate that there are meaningful differences in the modeled profiles across sites, or that any differences reflect the modeled differences in climate, vegetation or soil properties. Figures 3 and 4 demonstrate that the model can broadly fit a generic soil profile. However, it is unclear if the model can reliably capture differences between sites (for example, in Fig 3, the model reasonably fits only two of the four profiles). Comparison to a somewhat larger number of published soil F14C profiles is needed to support current statements that the model can "reproduce soil organic carbon stocks and radiocarbon profiles" (for example, line 29). This additional analysis would significantly strengthen the paper. It would also be particularly interesting to see if the model is able to capture the wide differences in bulk soil 14C seen across soil taxa (for example as explored in Mathieu et al (2015)).

Alternatively, if the authors feel that comparison to a wider suite of soil profiles is beyond the scope of the current work, the current model-data comparison should be rephrased as a proof-of-concept contribution. In either case, the discussion should address potential controls on the soil F14C profiles (for both data and model). For example, despite the important role of minerology and clay content in controlling the age of soil C, these topics are not mentioned in the current discussion. Relatedly, more discussion and exploration of the model processes and parameters that control the 14C profiles would be an important addition to this paper. Although I acknowledge that comparison to a wider suite of soil profiles may be beyond the scope of the current work, I would like to see more exploration and discussion of these issues prior to publication.

The authors make a good case for the addition of depth-varying parameters, both conceptually (eg line 69) and in the results, by making the important contribution of implementing He el al's suggested parameters in a depth-dependent context and updating the diffusion formulation. However, although the updated diffusion formulation is a key contribution of the paper, the impact of this model improvement should not be over-stated, as the difference between the two different model profiles relative to the data is not large (fig 3 &4). The modest gains suggest that adding other depth-varying processes in the future could be valuable. Although implementation of depth-varying parameters is clearly important, diffusion alone is not a singular model fix, and the discussion and conclusion should be broadened where possible to reflect this (for example, "mainly for diffusion" in line 40 and 468 is misleading/overstated).

I agree with Reviewer #1 on the major technical issue presented. This should be corrected prior to publication. The contribution of implementing the He et al (2016) suggested parameters is a good idea, and a nice contribution to the paper, so I would suggest retaining this model fit after updating the values as suggested by reviewer #1.

In general, figures could be made more professional, and a careful reading for grammatical errors is needed prior to publication.

In summary, this manuscript should be considered for publication after major revisions, including the technical fix presented by reviewer #1, model comparison to additional soil profiles, and/or an updated discussion of the results. Minor comments are listed below.

Specific comments:

Line 40 & 468: "mainly for diffusion" is misleading as discussed above

Lines 71-84: In introduction, cite other work using radiocarbon profiles to constrain soil models (e.g. Braakhekke et al, 2014; Ahrens et al, 2015)

Line 136-137: Please clarify, as this seems contradictory: "SOC diffusion is actually a representation of bioturbation processes (animal (and plant) activity), whereas DOC diffuses through concentration gradients." This text suggests that implementation of

SOC diffusion would not be based on a concentration gradient, while the Fick's law formulation provided (138-140) relies on a concentration gradient. Also, what do you mean by "the amount of carbon in the pool subject to transport"?

Line 181...: 14C data collection:

-Please clarify: was new data collected for this paper or is this published elsewhere?

-Please include a table of 14C data values, including sampling depth increments

-Please provide more methods details on soil collection and processing or reference to appropriate publication.

-How were litter and roots handed? Included/excluded? How does that correspond to model results?

Line 245-255: How are soil F14C values handled in the spinup? What is the potential influence on initial soil 14C values? Spinup is only $\sim$2 half lives of 14C and doesn't consider atmospheric variation prior to 1700.

Line 301: Please mention somewhere how comparisons are made between data and model, given differences in depths

Line 309-313 & Table 3: Visually, and discussed in the text, the sites Misiones and Feucherolles appear to have quite good fits for total soil carbon, while the fit is the worst for Mons, and also poor for Kissoko. However, the correlation coefficients are highest for Mons, but lowest for Kissoko. Is this a meaningful metric?

Table 3&4: Is there a reason all values have been rounded to end in .05 or .00 ?

Line 320-326/Fig 3: Any comments on why the model does so well in one French Luvisol (Feucherolles) and so poorly on the other (Mons) for total soil carbon? From the site description the sites sound very similar.

Line 334: "The vertical profile of the SOC stock simulated was thereby globally not very

far from that of the data". This seems like an overstatement based on results in Table 3. For example, although reported model total soil carbon is 1.7 and 2.1 overestimated at two sites with better fits, it is overestimated by a factor of 8.5 and 4.6 at the other two sites.

Fig 3: Relatedly, what depth ranges are used for comparison between data and model? How does this influence the results? For example, model and data look quite similar in Fig 3 for Misiones and Feucherolles, but the mean total soil carbon is reported to be overestimated by nearly a factor of 2.

Lines 364-366: Interesting, and nice to build on He et al (2016) using a depth-resolved approach

Line 392: More explanation of the results/implications of the priming effect mentioned here would be interesting, but not required

Lines 407-408: "Using a fixed diffusion constant implicitly suggests that soil fauna activity is uniform over the entire soil profile". Please add more explanation of the link between fauna activity and the diffusion term formulation for the reader. This diffusion term will vary with depth and across sites, because the Fick's law formulation also relies on the concentration gradient with depth. For example, in Kissoko, for much of the profile there is almost no change in total soil carbon with depth, so the diffusion term here would be zero. Does that imply that there is no soil fauna activity? Or simply that soil fauna activity does not result in a change in the soil carbon profile?

Lines 449-454: Well-stated summary of model contributions

Line 457: Please mention and cite any other land surface models that incorporate soil 14C either here or in introduction

Lines 466-468: "This suggests that, from now on, model improvements should mainly focus on a depth dependent parameterization, mainly for diffusion." Although diffusion did improve model results, the change was not dramatic. Please make sure the

language used here reflects the results.

-Broadly, figure aesthetics should be updated to look more professional throughout prior to publication. For example:

-Fig 7. Please label x & y axis. Please write depth increments for each bar on y-axis instead of 1-11. Also, in some of the panels numbers 11 and 12 are cutoff (eg 1..)

-Fig 3-7: Use more professional titles and punctuation on figures (eg. rather than "Model_Control" , "Model_Test He", etc.)

-Fig 7: It appears there are stray line numbers throughout the figures which will presumably be removed once the line numbers have been removed (eg fig 4,6,7)

-Update "litter structural below" and "litter metabolic below" to more clear and professional names

-Fig 7 is instructive and interesting. However, what is the reason for the "litter structural below" to decrease then increase again at the deepest depths in some of the profiles?

Language Comments: A careful and significant reading for grammatical errors and typos is needed prior to publication. A large number of very small changes are required. Here are a few examples (not comprehensive):

Line 59: "simulate" should be "simulates"

Line 71: typo "thIS"

Lines 74-77: very confusingly worded sentence

Line 81: "have" should be "has"

Line 84:"because of the conceptual description by pools non measurable" – fix grammar

Line 92: "yielded for the abrupt increase of atmospheric 14C concentration that doubles in 2-3 years." -clarify language

Line 198: "Congo Republic" should be "Republic of Congo"

Line 337: Missing period at end of sentence

Lines 659-660: "over the profile according to total soil carbon" - Meaning is unclear

Additional references:

Ahrens et al (2015). Contribution of sorption, DOC transport and microbial interactions to the 14C age of a soil organic carbon profile: Insights from a calibrated process model. Soil Biology and Biochemistry, 88. pp. 390-402.

Braakhekke et al (2014). The use of radiocarbon to constrain current and future soil organic matter turnover and transport in a temperate forest. Journal of Geophysical Research: Biogeosciences, 119(3).

Mathieu et al (2015). Deep soil carbon dynamics are driven more by soil type than by climate: a worldwide meta-analysis of radiocarbon profiles. Global Change Biology, 21. pp. 4278-4292.

---

## Referee Comment (RC3) · Anonymous Referee #3 · 23 Aug 2018

The cycling of organic matter through soil ecosystems is highly simplified in land surface models. This is a major source of uncertainty in projections of the terrestrial carbon sink under global climate change. Measurements of the radioactive carbon isotope 14C provides a powerful constraint for soil carbon models which include a radiocarbon tracer component. This manuscript documents the addition of a radiocarbon tracer component into the ORCHIDEE land model in order to enable radiocarbon constraints in it and in the IPSL Earth System Model it is coupled with. This study then demonstrated applying this constraint to the model based on several vertically-resolved soil radiocarbon profiles.

[Figure]

**General comments:**

The paper represents a substantial advance in the ORCHIDEE/IPSL model, which is an important tool in climate science, and has broader implications for other models. As such, it is well within the scope of GMD, and would represent a meaningful contribution to the field. However, there are several issues that would need to be addressed before I could recommend it for publication. I have detailed these issues below, and I hope that by addressing them, the authors will return with an improved presentation of this worthwhile research.

*Major issue 1:*

There are a couple of major issues with the Model_Test_He experiment. He *et al* (2006) suggested scaling the passive pool turnover time in IPSL/ORCHIDEE by 14, while scaling the slow-to-passive transfer coefficient by 0.07. I applaud the authors' effort to test this suggestion. However, the manuscript lacks a detailed explanation of exactly which quantities were scaled, and which of the arrows in Figure 1 corresponds with the first column of Table 2. The reduced complexity models of He *et al* consisted of three pools in series, whereas Figure 1 implies that ORCHIDEE has three soil pools that each independently exchange with a single pool of free DOC. Therefore, it seems that ORCHIDEE does not have a single transfer coefficient between slow and passive pools.

Furthermore, as pointed out in RC1, there seems to be an arithmetic error in the scaling of this transfer coefficient. The first and third rows of Table 2 imply that ORCHIDEE has some parameter with a value of 0.07 (this parameter being what needs improved explanation). Multiplying this by the scaling factor suggested in He *et al* would yield 0.0049, but it seems that 0.049 was used instead. The result is that the passive pool turnover time is increased by an order of magnitude without an equivalent adjustment to the inputs to this pool, leading to a large accumulation of radiocarbon-depleted SOM. This explains why the Model_Test_He experiment is so far off in Figures 3 and 5, and

why the standard bias is so high in Figures 4 and 6.

I would encourage the authors to re-run this experiment with the correct values and keep it in the manuscript (and, unlike RC1, I have no problem with the name). I understand that the recommended values were for a previous version of IPSL/ORCHIDEE, and that some of the changes since then (yielding ORCHIDEE-SOM, detailed in Camino-Serano *et al*, 2017) make the recommended changes superfluous by accounting for priming. Nevertheless, I think that testing these recommendations is a worthwhile exercise, even with this updated model version, and I would be interested in seeing it done correctly.

*Major issue 2:*

There is insufficient explanation of the depths at which the observational (field) data were sampled, and how that was compared with the model output. Figure 1 explains sufficiently the depth of the soil layers in the ORCHIDEE model (though an explanation in the main text would be welcome as well). The depth of the field measurements can be seen in Figures 3 and 5, but not with enough resolution to really understand. Was each field profile sampled at the exact same depths as the layers in ORCHIDEE, or is there some interpolation going on between one or the other?

The statistics in Section 2.6 are all over a dimension $i$, which I assume to represent the layers over depth, but this is not clearly stated. Given the importance of this $i$, we need more detail as to what it is. I would prefer to see an additional table or additional information in Table 1 to indicate how many samples were taken at each site and at what depths. And, most importantly, some explanation in the methods of how layer depths were harmonized between the model and observations, including an indication of the size of $i$ (i.e., the $n$ in the equations of Section 2.6).

Moreover, the specific depths at which the observed and modeled layers are compared should be clearly visible in Figures 3 and 5. The field observations are shown as points, with a single depth. Were measurements taken just at those single depths? Or

were entire layers sampled with an upper and lower boundary depth? The model is presumably providing an average concentration of carbon (and radiocarbon) for entire layers, but the lines in Figures 3 and 5 make it seem like the data are continuous rather than discrete.

Finally, the absence of explicit field data hinders the reproducibility of the study. The methods are described sufficiently to reproduce the study, and the model source code is available (though the web link has a problem, see below). But the study cannot be truly replicated without having access to the field data that were used. Including the field data in tabular format (perhaps as supplementary material) would go a long way toward making the methods more understandable and facilitating reproducibility.

*Major issue 3:*

THe authors provide some interpretation of each of the individual results in Section 3, but the manuscript lacks an overall discussion of the big-picture implications of these results and how they serve to advance scientific knowledge. The introduction section provides a compelling motivation for the study, but the manuscript lacks a sufficient discussion of how the current study informs these issues, what can be learned about SOM processes and soil-climate interactions, and what the implications are for the use of ESMs to project future climate change. I would like to see an expanded discussion of how these results fit in with the larger body of literature. The authors neglect to acknowledge that radiocarbon has already been implemented in a well known ESM (the Community Earth System Mode, CESM), and therefore do not discuss how their results relate to the existing work. The authors do cite the paper that would be relevant for this (Koven *et al*, 2013) in the context of diffusion representing bioturbation (line 406), but I would like to see an expanded discussion of how the results from the two papers potentially inform each other.

**Minor issues and technical corrections:**

Abbreviations: there are some abbreviations that are used without an explicit definition.

In some cases, they are defined later, but they should be defined in the first instance of use. I would avoid abbreviating SOC and SOM in the abstract, since neither one is used again in the abstract and just use the full text instead (but then define the abbreviation and begin using it when it first appears int he main body of the text). The abbreviation "F$^{14}$C" for fraction modern is used in the abstract, but not explicitly defined. "IPSL" is used several times before it is defined on line 105, and ORCHIDEE is never defined.

Line 71: spurious capitalization in the word "this"

Line 74: The sentence that begins on this line is too long, and should be broken up into at least two sentences to be understandable.

Line 75: "implementing" should be "to implement"

Lines 91-92: The decades should not have apostrophes (e.g., 1950s, not 1950's)

Line 93: Remove the word "since"

Line 94: Should be "As WITH any other carbon isotopes"

Lines 106–113: I am not sure how useful it is to list the names of the sub-components of ORCHIDEE without any further indication of how these components fit in to the present study. Instead, I would prefer to see a description of how ORCHIDEE fits into the larger ESM (e.g., which fluxes and state variables coupled it with the atmospheric model).

Line 158: There is some rendering issue with the $\delta$ (delta) symbol in $\delta^{13}$C; please double check.

Line 162: The abbreviations Asample and Aref should be explicitly defined for the sake of the reader who may be new to the concepts of radiocarbon.

Lines 167–179: There is some inconsistency between the main text and the equations regarding abbreviations. The text uses "$^{14}$C" while the equations use "$carbon14$". I believe these are supposed to represent the same thing, and should therefore have

the same abbreviations for clarity.

Lines 184–212: Some measurements include a space between the quantity and the units (e.g., "680 mm" on line 185) while others do not (e.g., "1.5m" on line 186)

Line 192: Define the abbreviation LSCE

Line 194: Define the abbreviation LMC14

Line 197: Define the abbreviation SOERE F-ORE-T

Line 232: The term "turnover rate" is ambiguous. I assume the authors mean "turnover time" since this is what He *et al* suggest should be scaled by 14, which would be the inverse of the decay "rate".

Line 252: What assumptions were made about the atmospheric $^{14}$C content during spinup?

Line 256: Were simulations actually run at a yearly time step? Section 2.1 indicates that some model components have a much shorter time step. Also, for comparison with the field data, was the final (2011) time step used?

Lines 339–340: Something is wrong with this sentence grammatically, which makes it difficult to interpret.

Lines 392–393: The 50

Line 408: Remove the word "fact" or add the word "in" before it.

Line 465-466: Please revise this sentence for grammatical accuracy.

Line 477: The provided website address links to a page that has issues with the SSL certificate, and will not load in any web browser without having to make a security exception. Providing the link as http rather than https would fix this issue, though the preferred solution would be maintain the https link and insure that the website has a valid SSL certificate.

---

## Author Comment (AC1) · 31 Oct 2018

Answer to comments from the reviewer #1.

We thank reviewer for the constructive evaluation of the manuscript. Please find below our answers to questions/comments.

Anonymous Referee #1

Improvement of the soil modules in global carbon cycle models is a recurrent need claimed by the scientific community. The land surface model Orchidee is one of the important tools to analyse and predict future changes of the Earth's climate and biosphere. A recent study highlighted that current Earth system models predict a too young age for soil organic carbon. The present work introduces the radiocarbon isotope in the model to better constrain Orchidee. Based on the use of radiocarbon, the study furthermore improved the model itself, and model prediction, through a better representation of carbon movement within the soil profile. The article is fully relevant, clearly written and illustrated, and worth publication in GMD.

ANSWER: Thank you very much for the positive comments

But one point requires a significant change. Once this point fixed, the paper could be acceptable with only minor corrections. Important point. (parameterization of the 'Model_Test_He') Line 233 authors state "... multiply by 14 the turnover rate and by 0.07 the flux...". And later Line 236 and in Table 2: "decrease the flux from 0.07 to 0.049. Consistency would either decrease the flux from 0.07 to 0.0049, or multiply it by a factor 0.7. I suppose that the initial intention of authors was to multiply by 0.07 the flux, so that the steady state stock of passive would be kept similar (multiplied by 14 x 0.07 = 0.98), but with a F14 much lower. Here it seems from the results that the stock of the passive pool was multiplied by a factor almost 10 (less than 10 because of the duration of the spin-up), as expected by a factor x 0.7 for the flux. The over estimation of both carbon content and age is obviously expected with such a parameterization. In the present state, I further recomment not to use the name of a person in the surname of the model. Finally, I recommend that the authors either (i) remove this model_test from the paper, which would then be accepted with minor revison, or (ii) recalculate using a flux to passive 0.0049 instead of 0.049. Option (ii) is preferred, but is not mandatory, since the other parts bring significant results; note that option (i) would not affect the summary nor the conclusion.

ANSWER: Actually this was only a typo mistakes in the manuscript but we carefully checked the code and it was correct. We therefore corrected the manuscript.

Minor and typographical points. Table 1. Be clear in the legend on what was averaged.

Do "over the profiles" means a calculated mean for (0- 2.0 m)"?

ANSWER: We did not have data up to 2m so we calculated a mean for the different available layers that we applied to the entire profile of the model (0-2.0 m). We clarified the legend: "Table 1. General description of the studied sites. The mean bulk density, pH and clay fraction values calculated from the different soil layers depths available in the data were used as input for each site. For Mons and Feucherolles sites, min and max values of pH and clay fraction are provided between brackets"

Table 3 Data are in kg C m-3, which is a unit for local concentration, not for carbon stock. Is it: (a) kg C m-2, i.e., the carbon stock per unit area; or (b) the average concentration over the 0-2.0 m profile (then the Stock would be 2 times the mean concentration value)? Option (a) would be preferred.

ANSWER: The table 3 was modified and all the results are now presented in kg C m-2.

Line 786 Legend fig.6: indicate the variable in object (= F14C).

ANSWER: This is now added in the legend

Line 134. A brief statement of the formalism and parameterization of the priming would be welcome.

ANSWER: We modified the text in the revised version as following: "Briefly, priming is described following equation 1 with DOCrecycled being the unrespired DOC that is redistributed into the pool i considered for each soil layer z in g C m-2 days-1, kSOC being a SOC decomposition rate constant (days-1), and LOC being the stock of labile organic C defined as the sum of the C pools with a higher decomposition rate than the pool considered within each soil layer z. We therefore considered that for the active carbon pool LOC is the litter and DOC, but for the slow carbon pool LOC is the sum of the litter, DOC and so on. Finally, c is a parameter controlling the impact of the LOC pool on the SOC mineralization rate, i.e., the priming effect. The equation was parameterized based on soil incubations data and evaluated over litter manipulation

experiments (Guenet et al. 2016)."

Line 179. In Eq (6), STRUC was excluded of total 14C. Was it? In the Century model, STRUC usually accounts for 10- 20% of C in 0-20 cm layer, and is therefore non negligible (your figure 7). It is considered as retrieved as material < 2mm (for a large part) and therefore often included in the "measured" total carbon. This exclusion may affect the comparison between observed and modelled values of F14.

ANSWER: In the model, structural litter may come from leaves or root litter production of the ongoing year. Soil scientists, before measurements, generally remove it. We agreed with the reviewer that a part can still be present after few years but we were not able to clearly define a time step when structural litter is less than 2mm and therefore integrated in the measurements. To avoid overestimation of modern C we decided to not integrate the structural litter in the final calculation. If needed we can perform a sensitivity analysis adding or not the structural litter in the final calculation to estimate the impacts on our results.

Line 218. "OCC (wt/wt)" would be better than "OCC (wt %)"

ANSWER: Done

Line 232. "turnover rate" is an unclear term (might be the reciprocal of turnover time). Here turnover time?

ANSWER: Correction is done in the revised version of the manuscript.

Lines 314-326 and throughout: MSD values aren't in kg C m-3, but in kg2 C m-6 (variance not standard deviation); or use squareroot(MSC)

ANSWER: We corrected the MSD units in the revised version of the manuscript.

Line 325. Arenosols are not very specific and are broadly represented on the planet. Remove "for such specific conditions". Replace by "Probably due to an overestimation of decay rates by ORCHIDEE in sandy soils?

ANSWER: This is now corrected in the revised version of the manuscript.

Lines 367-401. See Major comment.

ANSWER: As explained above, the error was in the table but not in the model.

Do is the boundary condition at depth 2.0 for constant diffusion affect the base of the profils?

ANSWER: It is difficult to answer this question because changing the soil depth of the model would not only affect the carbon but also the hydrology, the plant uptake and in fine the carbon inputs.

Typography

ANSWER: All the typo mistakes are corrected in the revised version. Line 71. "this" Line 158. verify in the final edition the greek symbol delta (not ok in my pdf) Line 186. "1.5_m" (= separate the units throughout) Line 227. "(2016) (=no square brackets) Line 242. "et al. (2014)" (spaces) Line 255. Point missing; also lines 326, 337 ... check. Line 315: spaces before and after "=" (throughout the text) Line 447 'processes
* * *

---

## Author Comment (AC2) · 31 Oct 2018

Answer to comments from the reviewer #2. We thank reviewer for the constructive evaluation of the manuscript. Please find below our answers to questions/comments.

Anonymous Referee #2

This paper presents ORCHIDEE-SOM-14C, a new version of the IPSL-Land Surface Model, and tests it against data from four different sites. It makes an important contribution by implementing the isotopic tracer 14C in the model. This is a valuable addition to the ORCHIDEE-SOM model, which simulates depth-resolved soil carbon dynamics

from 0-2m below the surface. The authors also demonstrate how the new model can be used to constrain SOC turnover times and internal model processes. In particular, they implement two variations on the model ("Model_Test_He" and "Model_Test_Diffusion"). They follow the suggestions of He et al (2016) to slow turnover in the passive pool and reduce the flux from the slow to passive pool (pending comment by reviewer #1). They also implement a version of the model with depth-dependent bioturbation rate following Jagercikova et al (2014). Conceptually, this paper is a nice demonstration of how F14C data could be used for comparison against different model implementations. However, there are significant issues which should be addressed both with the implementation (see Reviewer #1 comments) and interpretation of the results (see below) prior to publication.

ANSWER: Thanks for the positive comments please see our answer to reviewer #1.

In its current form, this paper does not convincingly demonstrate that there are meaningful differences in the modeled profiles across sites, or that any differences reflect the modeled differences in climate, vegetation or soil properties. Figures 3 and 4 demonstrate that the model can broadly fit a generic soil profile. However, it is unclear if the model can reliably capture differences between sites (for example, in Fig 3, the model reasonably fits only two of the four profiles). Comparison to a somewhat larger number of published soil F14C profiles is needed to support current statements that the model can "reproduce soil organic carbon stocks and radiocarbon profiles" (for example, line 29). This additional analysis would significantly strengthen the paper. It would also be particularly interesting to see if the model is able to capture the wide differences in bulk soil 14C seen across soil taxa (for example as explored in Mathieu et al (2015)). Alternatively, if the authors feel that comparison to a wider suite of soil profiles is beyond the scope of the current work, the current model-data comparison should be rephrased as a proof-of-concept contribution. In either case, the discussion should address potential controls on the soil F14C profiles (for both data and model). For example, despite the important role of minerology and clay content in controlling the age of soil C, these

topics are not mentioned in the current discussion. Relatedly, more discussion and exploration of the model processes and parameters that control the 14C profiles would be an important addition to this paper. Although I acknowledge that comparison to a wider suite of soil profiles may be beyond the scope of the current work, I would like to see more exploration and discussion of these issues prior to publication.

ANSWER: We agree that such an evaluation would be a good step forwards but one the difficulty to run the model over such a large database is that very often some boundaries conditions of the model are missing and we have to estimate them with large scale database that may not be accurate for a given site. In this study we decided to carefully choose some sites which have enough data to feed the model and which are also representative of different situations. We aimed to go for such large-scale evaluation but we thought that it would have been more useful to have first a model description papers evaluated on well-chosen site. We changed several parts of the document in the revised version of the manuscript to explore more this weakness of our study see for example: "Nevertheless, the model evaluation performed here on only four sites should be considered as proof of concept and more in depth evaluation are needed, in particular using a large 14C database available at global scale (Balesdent et al., 2018; Mathieu et al., 2015). Indeed, the F14C is largely controlled by pedo-climatic conditions such as clay content, climate and mineralogy (Mathieu et al., 2015) and the range of situations we covered here is relatively limited." or "Furthermore, here we used only one averaged value over the soil profile for soil boundary conditions (texture, pH, bulk density) but those variables are known to impact the F14C (Mathieu et al., 2015) and change with depth (Barré et al., 2009) and depth-varying boundary conditions may also help to improve the model.""

The authors make a good case for the addition of depth-varying parameters, both conceptually (eg line 69) and in the results, by making the important contribution of implementing He el al's suggested parameters in a depth-dependent context and updating the diffusion formulation. However, although the updated diffusion formulation is a key

contribution of the paper, the impact of this model improvement should not be over-stated, as the difference between the two different model profiles relative to the data is not large (fig 3 &4). The modest gains suggest that adding other depth-varying processes in the future could be valuable. Although implementation of depth-varying parameters is clearly important, diffusion alone is not a singular model fix, and the discussion and conclusion should be broadened where possible to reflect this (for example, "mainly for diffusion" in line 40 and 468 is misleading/overstated).

ANSWER: We agree with this statement and we add a paragraph in the conclusion to detail what should be the next step in the implementation of depth-varying parameters: "Here we presented the effect of a depth-varying diffusion constant but other parameters are depth dependent and should be represented in the next version of the model. For instance, belowground litter production in the model is simply represented by an exponential law without any representation of the effect of resource distribution on root profile (e.g. water or nutrients). This is a complex task in a land surface model aiming at running at large scale with a classical resolution of $0.5°$ but the soil modules of land surface models are quite sensitive to the NPP (Camino-Serrano et al., 2018; Todd-Brown et al., 2013) and a better constraint on the profile of the below ground litter production would probably improve the model performance."

I agree with Reviewer #1 on the major technical issue presented. This should be corrected prior to publication. The contribution of implementing the He et al (2016) suggested parameters is a good idea, and a nice contribution to the paper, so I would suggest retaining this model fit after updating the values as suggested by reviewer #1. In general, figures could be made more professional, and a careful reading for grammatical errors is needed prior to publication.

ANSWER: The error was a typo mistake in the manuscript but we carefully checked the code and it was correct.

In summary, this manuscript should be considered for publication after major revisions,

including the technical fix presented by reviewer #1, model comparison to additional soil profiles, and/or an updated discussion of the results. Minor comments are listed below.

Specific comments: Line 40 & 468: "mainly for diffusion" is misleading as discussed above

ANSWER: This is removed in the revised version.

Lines 71-84: In introduction, cite other work using radiocarbon profiles to constrain soil models (e.g. Braakhekke et al, 2014; Ahrens et al, 2015)

ANSWER: We added the citation the papers suggested by the reviewer: "Different authors have already succesfully implemented radiocarbon in soil models and were able to clearly show that the introduction of pools with turnover time of thousands of year were unnecessarry to fit radiocarbon data (Ahrens et al., 2015) whereas Braakhekke et al., (2014) showed that after a reparameterization of the models based on radiocarbon data the prediction of their model was quite different with more carbon in top soil and less in deep soil compared to the model without radiocarbon."

Line 136-137: Please clarify, as this seems contradictory: "SOC diffusion is actually a representation of bioturbation processes (animal (and plant) activity), whereas DOC diffuses through concentration gradients." This text suggests that implementation of SOC diffusion would not be based on a concentration gradient, while the Fick's law formulation provided (138-140) relies on a concentration gradient. Also, what do you mean by "the amount of carbon in the pool subject to transport"?

ANSWER: Both are based on a concentration gradient but the mechanisms we aimed to represent are different since it is bioturbation for the SOC whereas it is "real" diffusion for the DOC. We clarified the sentence: "SOC diffusion is actually a representation of bioturbation processes (animal (and plant) activity), whereas DOC relies more on a non-biological diffusion. Both diffuse through concentration gradients."

[Figure]

Line 181...: 14C data collection: -Please clarify: was new data collected for this paper or is this published elsewhere?

ANSWER: More details are now given see answer below.

-Please include a table of 14C data values, including sampling depth increments

ANSWER: We added the table 5 to present those data in the revised version of the manuscript.

-Please provide more methods details on soil collection and processing or reference to appropriate publication.

ANSWER: For the French sites information can be found in Jagercikova, M., S. Cornu, D. Bourlès, O. Evrard, C. Hatté, and J. Balesdent (2017), Quantification of vertical solid matter transfers in soils during pedogenesis by a multi-tracer approach, J. Soils Sediments, 17(2), 408–422, doi:10.1007/s11368-016-1560-9. The information is now added. For the two other sites, data are not published yet so we added more details. See for instance for the Misiones sites: "Details on measurements and sampling can be found in Tifafi et al., in prep. Briefly, the soil was sampled in May 2015 at different depth: 0-5cm, 5-10cm, 10-15cm, 15-20cm, 20-30cm, 30-40cm, 40-50cm, 50-60cm, 60-80cm, 80-100cm. All sampled were crushed and air-dried. Once in the laboratory, they were homogenized, crushed, randomly subsampled and sieved at 200$\mu$m. Then 14C measurements were made using a new Compact Radiocarbon System called ECHoMICADAS (Environment, Climate, Human, Mini Carbon Dating System) following the recommendation of Tisnérat-Laborde et al., (2015)."

-How were litter and roots handed? Included/excluded? How does that correspond to model results? ANSWER: Roots were removed when visible. In the model we used only the active, slow and passive pools to calculate the F14C but as mentioned by reviewer #1 structural litter might have been included in the calculation. Nevertheless, structural litter in the model can be part of the litter produced during the on-going year

but can also be few years old. Fix a threshold to determine which part of the structural litter would have been included needs underlying assumptions difficult to test. We therefore considered that only the soil carbon pools must be included in the calculation.

Line 245-255: How are soil F14C values handled in the spinup? What is the potential influence on initial soil 14C values? Spinup is only âĹij2 half-lives of 14C and doesn't consider atmospheric variation prior to 1700.

ANSWER: F14C were considered as stable before 1700. We considered this is a reasonable assumption since the variations observed from 1700 are mainly anthropogenic. The initialization procedure may indeed impact the results. If needed, we can perform a sensitivity analysis to the initial F14C.

Line 301: Please mention somewhere how comparisons are made between data and model, given differences in depths

ANSWER: We added this information: "The intervals of soil depth of the model outputs and the measurements were homogenized by interpolating linearly the data to common depth intervals defined for each site. The simulations and data were then compared for each depth interval."

Line 309-313 & Table 3: Visually, and discussed in the text, the sites Misiones and Feucherolles appear to have quite good fits for total soil carbon, while the fit is the worst for Mons, and also poor for Kissoko. However, the correlation coefficients are highest for Mons, but lowest for Kissoko. Is this a meaningful metric?

ANSWER: The good correlation coefficient for Mons is due to the relative good representation of the shape of the profile even though the mean bias is quite important as it is shown in Fig. 4. To clarify this point we added few words on this aspect at: "The correlation coefficient for Mons is relatively high compared to other site (Table 3) whereas Fig. 3 shows that the model performance was not very good for this site. This is mainly due to a large SB whereas other MSD components were rather low."

Table 3&4: Is there a reason all values have been rounded to end in .05 or .00?

ANSWER: It was pure random and following the recommendation of reviewer #1 we change the units of the total carbon from kg C m-3in kg C m-2 and the values from Table 3 do not all finished by .00 or .05

Line 320-326/Fig 3: Any comments on why the model does so well in one French Luvisol (Feucherolles) and so poorly on the other (Mons) for total soil carbon? From the site description the sites sound very similar.

ANSWER: This model like all the models following a similar structure are quite sensitive to the litter production. For Mons a net primary production (NPP) of 6.7 t ha-1 yr-1 was estimated by the technical institute for pasture in this region of France based on the annual yields, whereas the model predicts a NPP of 7.5 t ha-1 yr-1. The large over estimation might be a consequence of a bias in NPP. As far as we know no NPP estimation is available for Feucherolles. We added this information: "For Mons a net primary production (NPP) of 6.7 t ha-1 yr-1 was estimated by the technical institute for pasture in this region of France based on the annual yields, whereas the model predicts a NPP of 7.5 t ha-1 yr-1. The large overestimation of the SOC stocks may therefore be due to an overestimation of the NPP."

Line 334: "The vertical profile of the SOC stock simulated was thereby globally not very far from that of the data". This seems like an overstatement based on results in Table 3. For example, although reported model total soil carbon is 1.7 and 2.1 overestimated at two sites with better fits, it is overestimated by a factor of 8.5 and 4.6 at the other two sites. ANSWER: We rephrase to avoid overstatement. See line: "The vertical profiles of the SOC stock were fairly represented by the model" Fig 3: Relatedly, what depth ranges are used for comparison between data and model? How does this influence the results? For example, model and data look quite similar in Fig 3 for Misiones and Feucherolles, but the mean total soil carbon is reported to be overestimated by nearly a factor of 2.
ANSWER: This information is now added in the method section "The intervals of soil depth of the model outputs and the measurements were homogenized by interpolating linearly the data to common depth intervals defined for each site. The simulations and data were then compared for each depth interval."

Lines 364-366: Interesting, and nice to build on He et al (2016) using a depth-resolved approach

ANSWER: Thanks for the positive comments.

Line 392: More explanation of the results/implications of the priming effect mentioned here would be interesting, but not required

ANSWER: Since we did not run our model without priming we prefer to not increase the discussion section as it is to avoid over-interpretation.

Lines 407-408: "Using a fixed diffusion constant implicitly suggests that soil fauna activity is uniform over the entire soil profile". Please add more explanation of the link between fauna activity and the diffusion term formulation for the reader. This diffusion term will vary with depth and across sites, because the Fick's law formulation also relies on the concentration gradient with depth. For example, in Kissoko, for much of the profile there is almost no change in total soil carbon with depth, so the diffusion term here would be zero. Does that imply that there is no soil fauna activity? Or simply that soil fauna activity does not result in a change in the soil carbon profile?

ANSWER: Here we were wanted to talk about the diffusion rate and not the entire diffusion fluxes. We clarified the sentence: "Fick's law of diffusion is classically used in models to represent bioturbation assuming that soil fauna activity may be represented following the Fick's law of diffusion (Elzein and Balesdent, 1995; Guenet et al., 2013; Koven et al., 2013; O'Brien and Stout, 1978; Wynn et al., 2005). Using a fixed diffusion constant (D in eq. 2) implicitly suggests that soil fauna activity is uniform over the entire soil profile. This is generally the case of several models of diffusion especially used at

the level of an ecosystem (Bruun et al., 2007; Guimberteau et al., 2017; O'Brien and Stout, 1978). However, soil faunal activity vary naturally with depth and the diffusion constant should be depth-dependent (Jagercikova et al., 2014)."

Lines 449-454: Well-stated summary of model contributions

ANSWER: Thanks.

Line 457: Please mention and cite any other land surface models that incorporate soil 14C either here or in introduction.

ANSWER: We added a paper by Koven et al., 2013 in Biogeosciences.

Lines 466-468: "This suggests that, from now on, model improvements should mainly focus on a depth dependent parameterization, mainly for diffusion." Although diffusion did improve model results, the change was not dramatic. Please make sure the language used here reflects the results.

ANSWER: This was rephrased in the revised version

-Broadly, figure aesthetics should be updated to look more professional throughout prior to publication. For example: -Fig 7. Please label x & y axis. Please write depth increments for each bar on y-axis instead of 1-11. Also, in some of the panels numbers 11 and 12 are cutoff (eg 1..) -Fig 3-7: Use more professional titles and punctuation on figures (eg. rather than "Model_Control" , "Model_Test He", etc.) -Fig 7: It appears there are stray line numbers throughout the figures which will presumably be removed once the line numbers have been removed (eg fig 4,6,7) -Update "litter structural below" and "litter metabolic below" to more clear and professional names

ANSWER: All the figures have redo to more professional aspects.

-Fig 7 is instructive and interesting. However, what is the reason for the "litter structural below" to decrease then increase again at the deepest depths in some of the profiles?

ANSWER: The question might be that the diffusion constant D in deep layers has very

low values in deep soil because of the depth-varying equations we used. Therefore the diffusion fluxes are quite limited in deep layers. Furthermore, in deep soil the temperature is rather stable and those layers don't face important temperature increase in summer leading to high decomposition rates. Then, in deep soil the decomposition is limited and diffusion is not strong enough to homogenize the profile.

Language Comments: A careful and significant reading for grammatical errors and typos is needed prior to publication. A large number of very small changes are required.

ANSWER: All the grammatical errors were corrected and a native English speaker read the revised manuscript.

Here are a few examples (not comprehensive): Line 59: "simulate" should be "simulates" Line 71: typo "thIS" Lines 74-77: very confusingly worded sentence Line 81: "have" should be "has" Line 84:"because of the conceptual description by pools non measurable" – fix grammar Line 92: "yielded for the abrupt increase of atmospheric 14C concentration that doubles in 2-3 years." -clarify language Line 198: "Congo Republic" should be "Republic of Congo" Line 337: Missing period at end of sentence Lines 659-660: "over the profile according to total soil carbon" - Meaning is unclear

Additional references: Ahrens et al (2015). Contribution of sorption, DOC transport and microbial interactions to the 14C age of a soil organic carbon profile: Insights from a calibrated process model. Soil Biology and Biochemistry, 88. pp. 390-402. Braakhekke et al (2014). The use of radiocarbon to constrain current and future soil organic matter turnover and transport in a temperate forest. Journal of Geophysical Research: Biogeosciences, 119(3). Mathieu et al (2015). Deep soil carbon dynamics are driven more by soil type than by climate: a worldwide meta-analysis of radiocarbon profiles. Global Change Biology, 21. pp. 4278-4292.

---

## Author Comment (AC3) · 31 Oct 2018

Answer to comments from the reviewer #3.

We thank reviewer for the constructive evaluation of the manuscript. Please find below our answers to questions/comments.

Anonymous Referee #3

The cycling of organic matter through soil ecosystems is highly simplified in land surface models. This is a major source of uncertainty in projections of the terrestrial carbon sink under global climate change. Measurements of the radioactive carbon isotope

14C provides a powerful constraint for soil carbon models which include a radiocarbon tracer component. This manuscript documents the addition of a radiocarbon tracer component into the ORCHIDEE land model in order to enable radiocarbon constraints in it and in the IPSL Earth System Model it is coupled with. This study then demonstrated applying this constraint to the model based on several vertically-resolved soil radiocarbon profiles. General comments: The paper represents a substantial advance in the ORCHIDEE/IPSL model, which is an important tool in climate science, and has broader implications for other models. As such, it is well within the scope of GMD, and would represent a meaningful contribution to the field. However, there are several issues that would need to be addressed before I could recommend it for publication. I have detailed these issues below, and I hope that by addressing them, the authors will return with an improved presentation of this worthwhile research.

ANSWER: Thanks for the positive comments.

Major issue 1: There are a couple of major issues with the Model_Test_He experiment. He et al (2006) suggested scaling the passive pool turnover time in IPSL/ORCHIDEE by 14, while scaling the slow-to-passive transfer coefficient by 0.07. I applaud the authors'effort to test this suggestion. However, the manuscript lacks a detailed explanation of exactly which quantities were scaled, and which of the arrows in Figure 1 corresponds with the first column of Table 2. The reduced complexity models of He et al consisted of three pools in series, whereas Figure 1 implies that ORCHIDEE has three soil pools that each independently exchange with a single pool of free DOC. Therefore, it seems that ORCHIDEE does not have a single transfer coefficient between slow and passive pools.

ANSWER: To avoid making the manuscript too long we did not give all the details of the model construction (we mainly refer to Camino-Serrano et al. 2018) and mainly information on the 14C-related part. Nevertheless, in the model when the decomposed SOC goes to DOC, we keep track of the pool where it came from and the redistribution of the DOC once decomposed into SOC follows the same parameterization than the

ORCHIDEE version incorporated to the IPSL-ESM used by He et al., (2016). We therefore considered that using those parameter values still makes sense.

Furthermore, as pointed out in RC1, there seems to be an arithmetic error in the scaling of this transfer coefficient. The first and third rows of Table 2 imply that ORCHIDEE has some parameter with a value of 0.07 (this parameter being what needs improved explanation). Multiplying this by the scaling factor suggested in He et al would yield 0.0049, but it seems that 0.049 was used instead. The result is that the passive pool turnover time is increased by an order of magnitude without an equivalent adjustment to the inputs to this pool, leading to a large accumulation of radiocarbon-depleted SOM. This explains why the Model_Test_He experiment is so far off in Figures 3 and 5, and why the standard bias is so high in Figures 4 and 6. I would encourage the authors to re-run this experiment with the correct values and keep it in the manuscript (and, unlike RC1, I have no problem with the name). I understand that the recommended values were for a previous version of IPSL/ORCHIDEE, and that some of the changes since then (yielding ORCHIDEE-SOM, detailed in Camino-Serano et al , 2017) make the recommended changes superfluous by accounting for priming. Nevertheless, I think that testing these recommendations is a worthwhile exercise, even with this updated model version, and I would be interested in seeing it done correctly.

ANSWER: The error was a typo mistake in the manuscript but we carefully checked the code and we used the good value for the simulations.

Major issue 2: There is insufficient explanation of the depths at which the observational (field) data were sampled, and how that was compared with the model output. Figure 1 explains sufficiently the depth of the soil layers in the ORCHIDEE model (though an explanation in the main text would be welcome as well). The depth of the field measurements can be seen in Figures 3 and 5, but not with enough resolution to really understand. Was each field profile sampled at the exact same depths as the layers in ORCHIDEE, or is there some interpolation going on between one or the other?

[Figure]

ANSWER: More information is now given in the method section: "The intervals of soil depth of the model outputs and the measurements were homogenized by interpolating linearly the data to common depth intervals defined for each site. The simulations and data were then compared for each depth interval."

The statistics in Section 2.6 are all over a dimension i, which I assume to represent the layers over depth, but this is not clearly stated. Given the importance of this i, we need more detail as to what it is. ANSWER: We added this information: "x refers to the model outputs and y to the measurements, while i refers to soil depth."

I would prefer to see an additional table or additional information in Table 1 to indicate how many samples were taken at each site and at what depths. And, most importantly, some explanation in the methods of how layer depths were harmonized between the model and observations, including an indication of the size of I (i.e., the n in the equations of Section 2.6).

ANSWER: We added a table given the different layers and we add information on the interpolation method we used to compare data and model outputs (see previous answer).

Moreover, the specific depths at which the observed and modeled layers are compared should be clearly visible in Figures 3 and 5. The field observations are shown as points, with a single depth. Were measurements taken just at those single depths? Or were entire layers sampled with an upper and lower boundary depth? The model is presumably providing an average concentration of carbon (and radiocarbon) for entire layers, but the lines in Figures 3 and 5 make it seem like the data are continuous rather than discrete.

ANSWER: Since in the revised version of the manuscript we give both the interpolation method and the layers depth we hope that enough information is given to avoid misunderstanding.

Finally, the absence of explicit field data hinders the reproducibility of the study. The methods are described sufficiently to reproduce the study, and the model source code is available (though the web link has a problem, see below). But the study cannot be truly replicated without having access to the field data that were used. Including the field data in tabular format (perhaps as supplementary material) would go a long way toward making the methods more understandable and facilitating reproducibility.

ANSWER: Some of the data are already published and we refer now to the proper citation but others will be presented in an in prep. manuscript by the same authors. More details are now given. For instance: "Details on measurements and sampling can be found in Tifafi et al., In prep. Briefly, the soil was sampled in May 2015 at different depth: 0-5cm, 5-10cm, 10-15cm, 15-20cm, 20-30cm, 30-40cm, 40-50cm, 50-60cm, 60-80cm, 80-100cm. All sampled were crushed and air-dried. Once in the laboratory, they were homogenized, crushed, randomly subsampled and sieved at $200\mu$m. Then 14C measurements were made using a new Compact Radiocarbon System called ECHoMICADAS (Environment, Climate, Human, Mini Carbon Dating System) following the recommendations of Tisnérat-Laborde et al., (2015)."

Major issue 3: The authors provide some interpretation of each of the individual results in Section 3, but the manuscript lacks an overall discussion of the big-picture implications of these results and how they serve to advance scientific knowledge. The introduction section provides a compelling motivation for the study, but the manuscript lacks a sufficient discussion of how the current study informs these issues, what can be learned about SOM processes and soil-climate interactions, and what the implications are for the use of ESMs to project future climate change. I would like to see an expanded discussion of how these results fit in with the larger body of literature. The authors neglect to acknowledge that radiocarbon has already been implemented in a well known ESM (the Community Earth System Mode, CESM), and therefore do not discuss how their results relate to the existing work. The authors do cite the paper that would be relevant for this (Koven et al, 2013) in the context of diffusion representing bioturbation (line 406), but I would like to see an expanded discussion of how the results from the two papers potentially inform each other.

ANSWER: The paper by Koven et al., 2013 is now properly cited to its contribution "ORCHIDEE-SOM-14C, is one of the first land surface models that incorporates the 14C dynamic in the soil (Koven et al., 2013)." And more discussion has been added. For instance: "We limited our work here to depth-varying diffusion, but other parameters are also depth dependent and should be represented as such in the next version of the model. For instance, belowground litter production in the model is simply represented by an exponential law without any representation of the effect of resource distribution on root profile (e.g. water or nutrients). This is a complex task in a land surface model running at large scale with a classical resolution of 0.5°, but the soil modules of land surface models are quite sensitive to the NPP (Camino-Serrano et al., 2018; Todd-Brown et al., 2013) and a better constraint on the profile of the below ground litter production would likely improve the model performance. Furthermore, here we used only one averaged value over the soil profile for soil boundary conditions (texture, pH, bulk density) but those variables are known to impact the F14C (Mathieu et al., 2015) and change with depth (Barré et al., 2009) and depth-varying boundary conditions may also help to improve the model."

Or "Nevertheless, the model evaluation performed here on only four sites should be considered as proof of concept and more in depth evaluation are needed, in particular using a large 14C database available at global scale (Balesdent et al., 2018; Mathieu et al., 2015). Indeed, the F14C is largely controlled by pedo-climatic conditions such as clay content, climate and mineralogy (Mathieu et al., 2015) and the range of situations we covered here is relatively limited."

Minor issues and technical corrections:

ANSWER: All the minor issues and technical corrections were taken into account.

Abbreviations: there are some abbreviations that are used without an explicit definition.

In some cases, they are defined later, but they should be defined in the first instance of use. I would avoid abbreviating SOC and SOM in the abstract, since neither one is used again in the abstract and just use the full text instead (but then define the abbreviation and begin using it when it first appears int he main body of the text). The abbreviation "F14C" for fraction modern is used in the abstract, but not explicitly defined. "IPSL" is used several times before it is defined on line 105, and ORCHIDEE is never defined.

Line 71: spurious capitalization in the word "this" Line 74: The sentence that begins on this line is too long, and should be broken up into at least two sentences to be understandable. Line 75: "implementing" should be "to implement" Lines 91-92: The decades should not have apostrophes (e.g., 1950s, not 1950's) Line 93: Remove the word "since" Line 94: Should be "As WITH any other carbon isotopes" Lines 106–113: I am not sure how useful it is to list the names of the sub-components of ORCHIDEE without any further indication of how these components fit in to the present study. Instead, I would prefer to see a description of how ORCHIDEE fits into the larger ESM (e.g., which fluxes and state variables coupled it with the atmospheric model). Line 158: There is some rendering issue with the $\delta$ (delta) symbol in $\delta$13C; please double check. Line 162: The abbreviations Asample and Aref should be explicitly defined for the sake of the reader who may be new to the concepts of radiocarbon. Lines 167–179: There is some inconsistency between the main text and the equations regarding abbreviations. The text uses"14C" while the equations use "carbon14". I believe these are supposed to represent the same thing, and should therefore have the same abbreviations for clarity. Lines 184–212: Some measurements include a space between the quantity and the units (e.g., "680 mm" on line 185) while others do not (e.g., "1.5m" on line 186) Line 192: Define the abbreviation LSCE Line 194: Define the abbreviation LMC14 Line 197: Define the abbreviation SOERE F-ORE-T Line 232: The term "turnover rate" is ambiguous. I assume the authors mean "turnover time" since this is what He et al suggest should be scaled by 14, which would be the inverse of the decay "rate". Line 252: What assumptions were made about the atmospheric 14C content during spinup? Line 256: Were simulations actually run at a yearly time step? Section

2.1 indicates that some model components have a much shorter time step. Also, for comparison with the field data, was the final (2011) time step used? Lines 339–340: Something is wrong with this sentence grammatically, which makes it difficult to interpret. Lines 392–393: The 50 Line 408: Remove the word "fact" or add the word "in" before it. Line 465-466: Please revise this sentence for grammatical accuracy. Line 477: The provided website address links to a page that has issues with the SSL certificate, and will not load in any web browser without having to make a security exception. Providing the link as http rather than https would fix this issue, though the preferred solution would be maintain the https link and insure that the website has a valid SSL certificate.